# White matter structural bases for phase accuracy during tapping synchronization

**Pamela Garcia-Saldivar[1], Cynthia de León[1], Felipe A Mendez Salcido[1], Luis Concha[1,2]\*, Hugo Merchant[1]\***

[1]Institute of Neurobiology, Universidad Nacional Autónoma de México, Campus Juriquilla, Querétaro, Mexico; [2]International Laboratory for Brain, Music and Sound (BRAMS), Montreal, Canada

**Abstract** We determined the intersubject association between the rhythmic entrainment abilities of human subjects during a synchronization-continuation tapping task (SCT) and the macro- and microstructural properties of their superficial (SWM) and deep (DWM) white matter. Diffusion-weighted images were obtained from 32 subjects who performed the SCT with auditory or visual metronomes and five tempos ranging from 550 to 950 ms. We developed a method to determine the density of short-range fibers that run underneath the cortical mantle, interconnecting nearby cortical regions (U-fibers). Notably, individual differences in the density of U-fibers in the right audio-motor system were correlated with the degree of phase accuracy between the stimuli and taps across subjects. These correlations were specific to the synchronization epoch with auditory metronomes and tempos around 1.5 Hz. In addition, a significant association was found between phase accuracy and the density and bundle diameter of the corpus callosum (CC), forming an interval-selective map where short and long intervals were behaviorally correlated with the anterior and posterior portions of the CC. These findings suggest that the structural properties of the SWM and DWM in the audiomotor system support the tapping synchronization abilities of subjects, as cortical U-fiber density is linked to the preferred tapping tempo and the bundle properties of the CC define an interval-selective topography.

**\*For correspondence:**
lconcha@unam.mx (LC);
hugomerchant@unam.mx (HM)

## Editor's evaluation

This paper is valuable in that it provides a critical missing link between measures of structural connectivity and rhythmic tapping abilities, pointing to interesting possibilities for how tapping synchronization is carried out. The methodology and findings are convincing, and of interest to those studying the neural mechanisms of timing.

## Introduction

Moving in synchrony with regular musical events (i.e., beat) is a basic and generalized human ability that can reach sophisticated levels in professional percussionists (*Honing et al., 2015*; *Merchant et al., 2015b*). Indeed, humans are extremely sensitive to auditory regularities and can entrain to auditory beats across a wide range of tempos, as well as use timed movements of different body parts (such as finger or foot taps or body swaying) to keep the beat (*Mendoza and Merchant, 2014*; *Repp and Su, 2013*). A classical task used to study rhythmic entrainment is the synchronization-continuation task (SCT), where subjects first entrain their tapping to a set of isochronous stimuli, also known as a metronome, and then continue tapping without the periodic stimulus using an internal clock (*Merchant et al., 2008a*; *Wing, 2002*; *Wing and Kristofferson, 1973*). In this type of task, humans show negative asynchronies, namely they tap a few milliseconds before the metronome,

supporting the notion that entrainment depends on a predictive internal beat that is phase-locked to the stimuli (*Lenc et al., 2021*; *Repp and Su, 2013*; *Zarco et al., 2009*). Neurophysiological and functional imaging studies have shown that the internal beat representation during an SCT rests in the motor system, including the basal ganglia and supplementary motor regions (*Bartolo et al., 2014*; *Merchant et al., 2013a*; *Merchant and Averbeck, 2017*; *Rao et al., 2001*; *Sánchez-Moncada et al., 2024*). These areas produce a regenerating rhythmic signal (*Crowe et al., 2014*; *Gámez et al., 2019*; *Merchant et al., 2015a*) that dynamically interacts with the auditory areas, creating audiomotor loops where the motor prediction of the beat is flexibly compared and adjusted depending on changes in the input stream of rhythmic stimuli to phase-lock sensory and motor signals (*Comstock et al., 2018*; *Merchant and Honing, 2014b*; *Patel and Iversen, 2014*). In addition, neurons in the supplementary motor areas (SMAs) are tuned to the tempo duration of tapping (*Merchant et al., 2014a*; *Merchant et al., 2013b*), giving rise to interval-specific circuits that define a chronotopic map with short preferred intervals in the anterior portion and long preferred intervals in the posterior portion of the medial premotor areas (*Protopapa et al., 2019*; *Merchant et al., 2024b*).

The existence of neural circuits with preferred intervals comprising chronotopic maps is consistent with the human flexibility to tap in phase (with asynchronies close to zero) and high precision to isochronous stimuli over a wide range of interstimulus-onset intervals (ISIs), spanning from 400 to 1200 ms (*Mates, 1994*; *Repp, 2005*). Within this window, subjects demonstrate a spontaneous rhythmic tempo, which corresponds to the interval produced naturally when asked to tap in without external cues (*McAuley et al., 2006*; *Zamm et al., 2018*). This spontaneous or preferred tempo is around 600–750 ms in human adults (*Fischl, 2012*; *Fraisse, 1978*, but see *Parncutt, 1994*), but is faster in early childhood and slower in late adulthood (*McAuley et al., 2006*). A recent study demonstrated that the perception of rhythmic stimuli also has a preferred tempo, with an optimal sampling rate of ~1.4 Hz (ISI of 714 ms) in audition and ~0.7 Hz (ISI of 1428 ms) in vision. Furthermore, motor tapping helps to synchronize the temporal fluctuations of attention with maximal effects at ~1.7 Hz (ISI of 588 ms), but only for the auditory modality (*Zalta et al., 2020*). These findings support the notion that ongoing motor activity shapes attention and beat perception, as it imposes temporal constraints on the sampling of sensory information within a narrow frequency range (*Morillon et al., 2019*). Hence, the audiomotor system is built to optimally work at a preferred tempo.

Although rhythmic entrainment is prevalent across all human cultures and is a natural behavior for social interaction (*Nettle, 2000*; *Jacoby et al., 2024*), there are wide individual differences in the period (inter-tap interval) and phase (asynchronies) of movement synchronization. There are subjects who lack musical training, yet spontaneously synchronize to rhythmic stimuli ranging from strictly periodic metronomes to complex musical pieces, with performance comparable to that of trained musicians (*Scheurich et al., 2018*; *Tranchant et al., 2016*). Conversely, there are poor synchronizers (around 10% of the population) that show low period accuracy and large asynchronies to isochronous metronomes and musical excerpts (*Phillips-Silver et al., 2011*; *Sowiński and Dalla Bella, 2013*; *Tranchant et al., 2016*). Furthermore, non-musicians synchronize less flexibly and less precisely across tempos than musicians (*Scheurich et al., 2018*). Hence, both genetic and learning factors influence the beat-entrainment abilities of humans. A large-scale genome-wide association study (GWAS) recently demonstrated a highly polygenic architecture of the human capacity to synchronize to a musical beat. The GWAS phenotype for beat synchronization was related to performance in beat synchronization tasks and rhythm perception tasks (*Niarchou et al., 2022*). Nevertheless, genetic influences account for only a small portion of human variation in beat synchronization, while environmental influences are the primary drivers of rhythmic accuracy. In fact, functional imaging has revealed that individual differences in beat perception depend on activation differences in the SMA and the posterior auditory cortex (*Grahn and McAuley, 2009*). In addition, the putamen, SMA, and auditory cortex show greater functional connectivity during rhythm perception, with larger modulation for musicians than non-musicians (*Grahn and Rowe, 2009*; *Grahn and Rowe, 2013*). Consequently, the magnitude of the anatomofunctional association between the auditory and motor control areas of the cerebral cortex seems to covary with the individual difference in how humans perceive and entrain to simple regular beats. Such brain networks rely on the structural connectivity provided by white matter, which can be evaluated through diffusion-weighted imaging (DWI). Long-range anatomical connectivity is supported by deep white matter (DWM) bundles, while short-range connectivity is achieved through fibers that run tangentially to the cortical surface and connect adjacent and proximal cortical

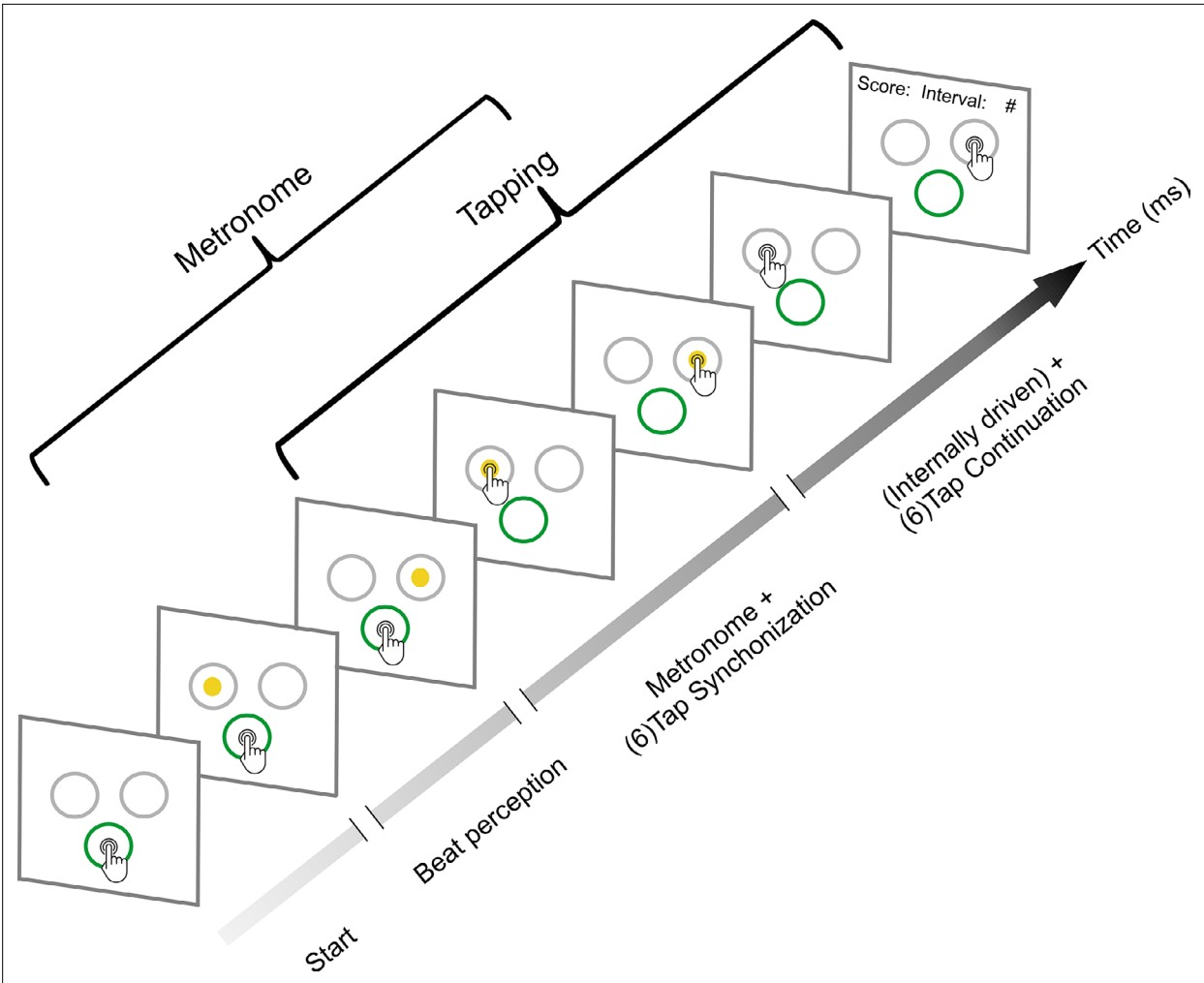

**Figure 1.** Synchronization-continuation task (SCT). Initially, the subjects placed their finger at the central-bottom target of a touchscreen to start the trial and maintained the finger in this position while observing a sequence of right–left alternating visual stimuli with a constant interstimulus interval (target duration, perception epoch). The subjects were instructed to start tapping once they got the beat from the metronome; they had two to six stimuli to start tapping. Thus, when they considered they had extracted the beat, subjects tapped the touchscreen on the left or right target in synchrony with the alternating visual metronome for six intervals (synchronization epoch). Finally, they continued tapping on the right or left targets for another six intervals without the metronome (continuation epoch). The mean produced and the target interval were displayed at the end of each trial as feedback. Subjects also performed an auditory metronome version of the SCT, where the metronome consisted of 500 Hz tones that were presented on the right or left side of a headphone. Five target durations (550, 650, 750, 850, or 950 ms) were presented pseudorandomly, with the visual and auditory conditions interleaved between subjects.

regions. These fibers are collectively termed U-fibers due to their shape (*Schüz and Braitenberg, 2002*; *Schmahmann et al., 2006*, *Shastin et al., 2022*). In this work, we individually analyze the role of these two forms of connectivity in rhythmic entrainment abilities. We hypothesize that if the cortical connectivity of the audiomotor system is defining rhythmic entrainment abilities, then individual differences in tapping synchronization should covary with the degree of anatomical connectivity (*Assaneo et al., 2019*; *Steele et al., 2013*). Previous evidence suggests that the audiomotor system is tuned at a limited interaction rate (*Zalta et al., 2020*; *Morillon et al., 2019*). Hence, we also hypothesize that the relationship between rhythmic tapping abilities and the structural connectivity of the audiomotor system should be more evident for intervals close to the preferred tempo.

To test these hypotheses, we acquired DWIs from 32 subjects that had previously performed an SCT using flashing visual or auditory tones as metronomes in the range of hundreds of milliseconds (*Figure 1*). SCT rhythmic performance across durations (ISI: 550, 650, 750, 850, or 950 ms) and modalities (auditory and visual) was characterized using the absolute asynchronies, the autocorrelation

of the inter-tap interval time series during the synchronization epoch, the constant error, and the temporal variability during both synchronization and continuation epochs. These parameters measure the phase accuracy, error correction, period accuracy, and period precision of the rhythmic tapping of the subjects, respectively (*Figure 2A*). We used an ISI range of 550–950 ms because it contains the preferred interval and is within the optimal window for tap synchronization (*Repp, 2005*). Hence, with this ISI range, we could potentially identify structural correlates for both the preferred tempo and interval selectivity. With this in mind, we evaluated DWM fascicles using a fixel-based approach (*Dhollander et al., 2021*) and developed two metrics for superficial white matter (SWM): fiber density corresponding to fibers entering or exiting the cortex and U-fibers running tangentially to the cortex. Widespread correlations in the right audiomotor circuit were found between the tangential U-fiber density and the phase accuracy of subjects during the synchronization epoch of the auditory condition for the 650 and 750 ms intervals. The interval specificity in these associations suggests that the preferred tempo for rhythmic entrainment has its origins in the structural properties of the U-fibers running superficially across the audiomotor circuit. In addition, there was a significant association between asynchronies in the auditory condition and the density and bundle diameter of the corpus callosum (CC), forming an interval-selective map with an anterior–posterior gradient, similar to the topography of interval-tuned clusters observed with functional imaging. Crucially, the anatomo-behavioral associations were negative, indicating that subjects with good predictive abilities and small asynchronies exhibited large superficial and deep apparent fiber densities (AFDs), while subjects with large asynchronies showed low fiber densities.

## Results

### Rhythmic behavior

Thirty-two subjects performed a modified version of the classical SCT that included the following three epochs: beat perception, synchronization, and continuation (*Figure 1*). This task starts with the active perception of the isochronous beat defined by alternating left–right visual stimuli, followed by tapping synchronization to the alternating stimuli, and the internally driven tapping continuation to the right or left targets without the metronome (*Pérez et al., 2023*). The subjects also performed an auditory version of the SCT (*Figure 1*, see Methods).

Absolute asynchronies correspond to the time difference between each stimulus and response pair and are a measure of the phase accuracy between taps and stimuli (*Figure 2A*). Hence, this parameter can only be measured during the synchronization epoch of the SCT. We performed a repeated-measures analysis of variance (ANOVA) on absolute asynchronies with metronome modality (auditory and visual: two levels) and instructed interval duration (550, 650, 750, 850, and 950 ms: five levels) as within-subject factors. The ANOVA showed significant main effects for duration ($F(4,124)$ = 36.88, p < 0.0001) and modality ($F(1,31)$ = 20, p < 0.0001), as well as a significant duration × modality interaction ($F(4,124)$ = 32.3, p < 0.0001). Tukey's honest significant difference (HSD) post hoc test showed significantly larger asynchronies for the visual than auditory modality across all durations (p < 0.0001 for 550, 650, 750, and 850 ms; p = 0.006 for 950 ms; with the interaction effect mainly driven by the difference in the 950 ms across modalities). These results confirm the preponderance of the auditory modality over the visual modality to produce phase alignment of the taps with the metronome (*Comstock et al., 2018*; *Gámez et al., 2018*; *Merchant et al., 2015a*). In addition, we computed the intersubject correlation matrix on the absolute asynchronies across instructed intervals and found a significant correlation between 650 and 750 ms for both the auditory and visual conditions ($r$ = 0.67, p = 0.000026; $r$ = 0.69, p = 0.000011, respectively) (*Figure 2—figure supplement 1*). This finding suggests the existence of a shared mechanism for metronome-tap phase alignment in the intervals that correspond to the preferred tempo indicated in previous studies, thereby corroborating the notion that the audiomotor system is efficiently tuned to this tempo (*Zalta et al., 2020*).

A negative lag 1 autocorrelation of the produced intervals during the synchronization epoch indicates the involvement of an error correction mechanism that maintains tap synchronization with the metronome, since a longer produced interval tends to be followed by a shorter interval, while a shorter interval tends to be followed by a longer produced duration (*Figure 2A*; *Iversen et al., 2015*; *Repp and Su, 2013*). The corresponding repeated-measures ANOVA on autocorrelation of the inter-tap interval time series (*Figure 2C*) showed no significant differences between modalities ($F(1,31)$ = 1.6, p

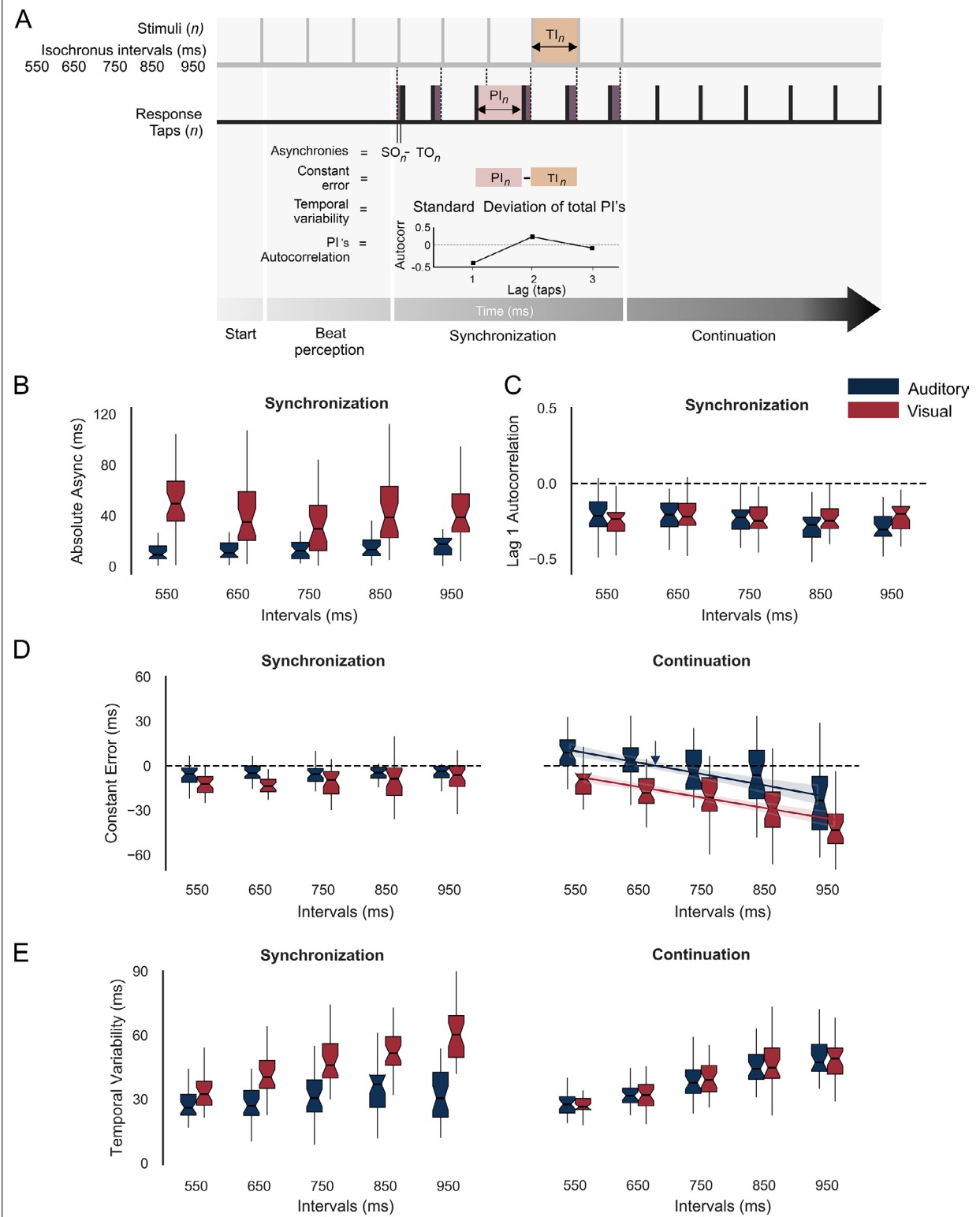

**Figure 2.** Behavior during the synchronization-continuation task (SCT). (**A**) Parameters of rhythmic performance during the SCT. Asynchronies correspond to the time difference between stimulus (SO) and tap onset (TO) across the *n* intervals of the synchronization epoch. The constant error is the difference between produced (PI) and target intervals (TI), and the temporal variability is the standard deviation of the PI. Finally, the autocorrelation of the PI during synchronization and the lag 1 autocorrelation are computed. A negative value indicates that the subject is using an error correction mechanism (see the text). (**B**) Absolute asynchronies for each instructed interval and metronome modality (auditory: blue, visual: red) during the

*Figure 2 continued on next page*

Figure 2 continued

synchronization epoch. (**C**) Lag 1 autocorrelation for each interval and modality during the synchronization epoch. (**D**) Constant error as a function of target interval for both metronome modalities and the synchronization (left) and continuation (right) epochs of the SCT. The colored lines in D for the continuation epoch correspond to the linear fit between the constant error and the target interval; the indifference interval corresponds to 680 ms (blue vertical arrow) for the auditory condition. (**E**) Temporal variability as a function of target interval for both metronome modalities and the two epochs of the SCT.

The online version of this article includes the following figure supplement(s) for figure 2:

**Figure supplement 1.** Correlation matrix for the intersubject absolute asynchronies across instructed intervals for the auditory (left) and visual (right) conditions.

= 0.21), intervals ($F_{(4,124)}$ = 2.05, p = 0.09) or their interaction ($F_{(4,124)}$ = 2.34, p = 0.06). Thus, lag 1 autocorrelation across trials was negative (~80% of the trials) and similar across modalities and target durations, supporting the notion of a robust and amodal error correction mechanism during the SCT.

Constant error is the difference between produced and target intervals and is a measure of period accuracy during the synchronization and continuation epochs (*Figure 2A*). A repeated-measures ANOVA on constant error with modality, target duration, and task epoch as within-subject factors revealed statistically significant main effects for modality ($F_{(1,31)}$ = 46.05, p < 0.0001), target duration ($F_{(4,124)}$ = 29.15, p < 0.0001), and epoch ($F_{(1,31)}$ = 6.44, p = 0.01), as well as significant interactions between modality × interval ($F_{(4,124)}$ = 3.5, p = 0.008), epoch × interval ($F_{(4,124)}$ = 48.33, p < 0.0001), and epoch × modality ($F_{(1,31)}$ = 23.35, p < 0.0001). The post hoc Tukey HSD showed no significant differences for target duration in constant error during synchronization for both modalities, with accurate timing close to zero. In contrast, during continuation, the same post hoc test revealed that the significant interactions between factors were mainly due to the significant differences between distant intervals within the auditory and visual modalities (*Figure 2D*). In fact, for the continuation epoch, the constant error followed the bias effect, with overestimation for short durations and underestimation for long durations, especially for the auditory condition (*Jazayeri and Shadlen, 2010*; *Pérez and Merchant, 2018*; *Pérez et al., 2023*). Indeed, the indifference interval, which corresponds to the interval associated with zero constant error, was 654 ms for the auditory condition (*Figure 2D*, blue arrow) and 420 ms for the visual condition. This finding suggests that our subjects had a clear preferred interval in the auditory condition that is close to the 2 Hz reported in the literature (*Zamm et al., 2016*). Finally, temporal variability was defined as the standard deviation of the produced intervals and is a metric of timing period precision (*Figure 2A*). The same repeated-measures ANOVA on temporal variability showed significant main effects for target duration ($F_{(4,124)}$ = 110, p < 0.0001) and modality ($F_{(1,31)}$ = 58.06, p < 0.0001), but no significant main effect for task epoch ($F_{(1,31)}$ = 0.94, p = 0.33). In addition, significant effects were revealed for the following interactions: epoch-interval ($F_{(4,124)}$ = 3.42, p = 0.01), epoch-modality ($F_{(1,31)}$ = 70.68, p < 0.0001), interval-modality ($F_{(4,124)}$ = 11.34, p = 0.0001), and epoch-modality-interval ($F_{(4,124)}$ = 14.74, p = 0.0001). The HSD post hoc tests showed significantly greater temporal variability in the visual metronome than in the auditory metronome (for the intervals 650, 750, 850, and 950 ms: p < 0.0001) during the synchronization but not the continuation epoch, confirming the high period precision for auditory metronomes, especially during synchronization (*Figure 2E*; *Gámez et al., 2018*; *Repp and Penel, 2004*).

## White matter analysis

Before the task performance session, participants were scanned in a 3T Philips Achieva TX MR scanner with a 32-channel head coil. T1-weighted volumes and DWIs were obtained (see Methods). For each subject, the gray/white matter interface was identified using a surface mesh (*Fischl, 2012*; *Fischl et al., 2002*). The AFD (*Raffelt et al., 2012*), a metric that non-invasively evaluates axonal density (*Rojas-Vite et al., 2019*), was derived from DWI using constrained spherical deconvolution (CSD) (*Tournier et al., 2004*), and sampled at each vertex of this mesh. To evaluate white matter properties at different depths with respect to the cortical mantle, we created synthetic trajectories that organically extended from each vertex at the gray/white matter interface toward the ventricles and sampled diffusion metrics along these trajectories every 0.5 mm (*Figure 3A*). Leveraging the ability of CSD to disentangle crossing fiber populations, AFD (*Figure 3B*) was evaluated separately for those fibers that enter or exit the cortex and are, therefore, parallel to the virtual trajectories ($_{par}$AFD), and those that extend tangentially to the cortex, perpendicular to the virtual trajectories ($_{tan}$AFD). Throughout this

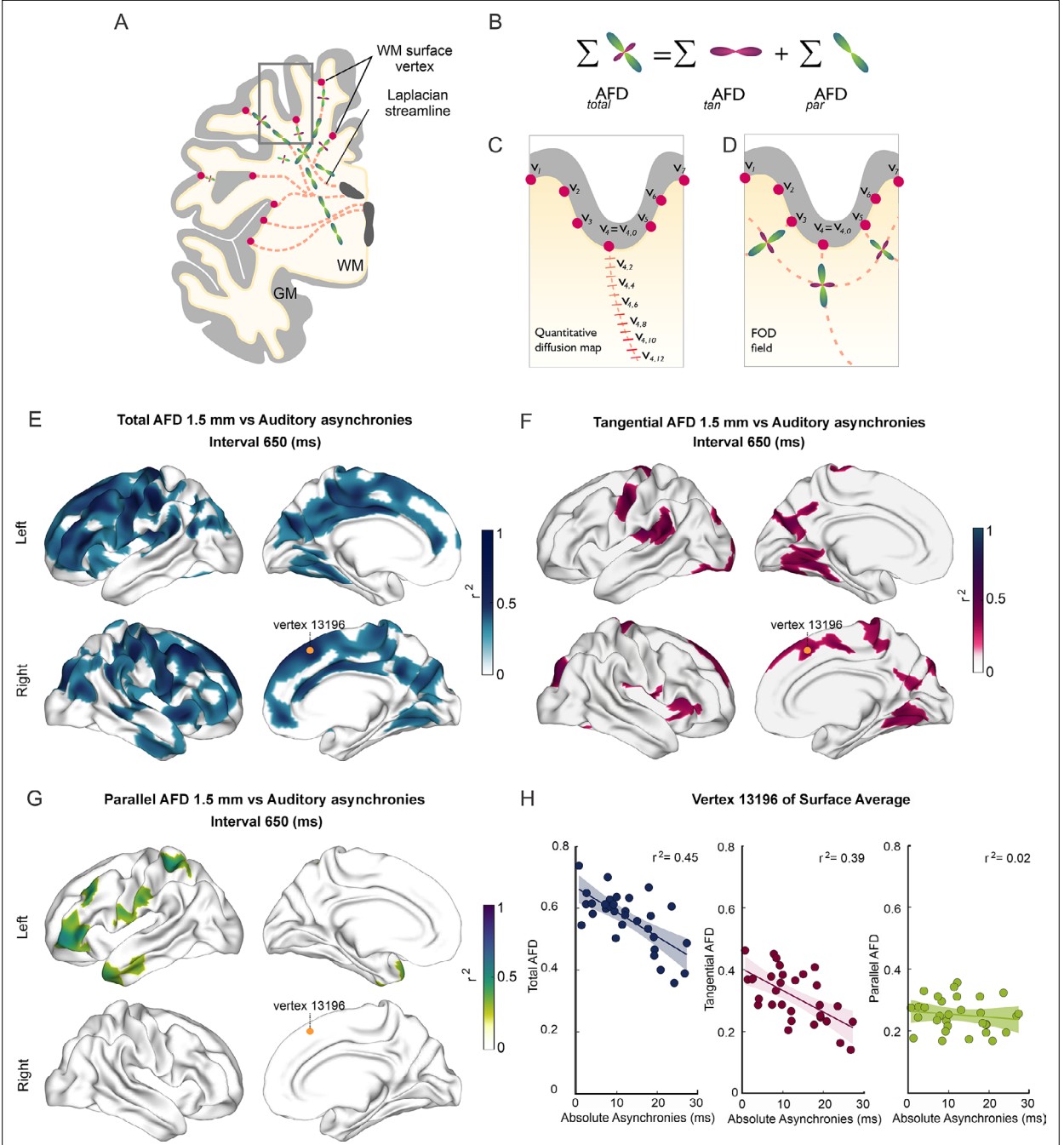

**Figure 3.** Superficial white matter analysis. (**A–D**) Apparent fiber density (AFD) of superficial white matter was systematically sampled (red dashes in C) along synthetic streamlines (red lines in A and C) that extend from each vertex (red circles) of the gray/white matter interface surface toward the ventricles following a Laplacian field (**A–C**). The integral of all fiber orientation distribution functions (FOD, D) corresponds to the total apparent fiber density ($_{total}$AFD), further separated (**B**) into fiber densities corresponding to fibers entering/exiting the cortex parallel to the Laplacian streamlines ($_{par}$AFD) and U-fibers running tangentially to the cortex ($_{tan}$AFD). (**E**) There are widespread significant correlations between the subjects' asynchronies during the synchronization-continuation task (SCT) with an auditory metronome and $_{total}$AFD, shown here for the 650 ms interval (**E**). (**F**) Large areas within the frontal, parietal, and occipital lobes showed significant correlations between behavior and $_{tan}$AFD. (**G**) Only restricted frontal and temporal regions showed correlation between $_{par}$AFD and the asynchronies. (**H**) Coefficient of determination across the 32 subjects between the three AFD metrics and asynchronies for one exemplary vertex (yellow dot in E–G).

The online version of this article includes the following video and figure supplement(s) for figure 3:

**Figure supplement 1.** Characterization of the superficial white matter (SWM) properties associated with synchronization-continuation task (SCT) performance.

*Figure 3 continued on next page*

Figure 3 continued

**Figure 3—video 1.** Spatial correlations between the superficial white matter (SWM) apparent fiber density and interval-specific absolute asynchronies of the auditory modality.

https://elifesciences.org/articles/83838/figures#fig3video1

**Figure 3—video 2.** Spatial correlations between the tangential superficial white matter (SWM) apparent fiber density and interval-specific absolute asynchronies of the auditory modality.

https://elifesciences.org/articles/83838/figures#fig3video2

**Figure 3—video 3.** Spatial correlations between the parallel superficial white matter (SWM) apparent fiber density and interval-specific absolute asynchronies of the auditory modality.

https://elifesciences.org/articles/83838/figures#fig3video3

work, we assume that $_{par}$AFD is related to association, commissural, and projection fibers that eventually enter or exit DWM bundles, while $_{tan}$AFD is informative of short-range cortico-cortical connections through U-fibers (*Schüz and Braitenberg, 2002*).

## Correlations between behavior and SWM

Next, we examined the association between behavioral performance and the microstructural properties of SWM inferred from DWIs. The surface-based analysis of SWM (see Methods) was performed to determine the possible association between the different metrics of rhythmic timing performance and the metrics of the SWM sampled at five depths with respect to the gray/white matter interface (0, 0.5, 1, 1.5, and 2 mm). This analysis showed the existence of negative correlations between the auditory absolute asynchronies and values of the AFD maps in the five depths sampled (*Figure 3E–G*; also see *Figure 3—figure supplement 1*). Thus, subjects with auditory asynchronies closer to zero, and hence with larger predictive abilities, had significantly higher AFD and $_{tan}$AFD values than subjects with less predictive performance.

Notably, these significant associations were observed mainly for the auditory asynchronies of the 650 and 750 ms intervals and the $_{total}$AFD and $_{tan}$AFD values (see *Figure 3—videos 1–3*). No significant correlations were found between the three AFD maps and the constant error, temporal variability, and lag 1 autocorrelation for the auditory condition. Furthermore, no correlations were observed between the three metrics of the SWM and all the behavioral parameters for the visual condition (see *Figure 3—figure supplement 1*). We did not find significant correlations between the parameters of SCT rhythmic performance and the $_{par}$AFD, except for a few clusters with low correlation coefficients for the auditory asynchronies at the 550, 650, and 750 ms intervals (*Figure 3—figure supplement 1* and *Figure 3H*). Indeed, the level of association between SCT phase accuracy and the AFD maps was greater for $_{tan}$AFD than for $_{par}$AFD (*Figure 3H*; compare Figure 5 with *Figure 5—figure supplement 1*).

According to *Schüz and Braitenberg, 2002*, the average depth of the U-fiber system is approximately 1.5 mm. Thus, the following analyses were done at this depth. *Figure 4* shows the correlation coefficient of determination values between the auditory asynchronies and $_{tan}$AFD at 1.5 mm below the gray/white matter interface across all the tested tempos. After correction for multiple comparisons (pcft < 0.001 and $p_{cluster}$ < 0.001), only intervals of 650, 750, and 850 ms showed a significant association between the behavioral and structural parameters. Indeed, nine (with 2232 vertices), twelve (with 2827 vertices), and two (with 374 vertices) clusters showed significant correlations between auditory asynchronies and $_{tan}$AFD for the 650, 750, and 850 ms intervals, respectively. *Figure 4—figure supplement 1* shows the association between asynchronies (650 ms intervals) and $_{tan}$AFD at various depths from the gray/white matter interface.

To identify the anatomical regions with significant clusters of vertices, we parcelled the SWM based on the Brainnetome Atlas (*Fan et al., 2016*). Areas with significant vertices (after correction for multiple comparisons; pcft < 0.001 and $p_{cluster}$ < 0.001) were grouped into 14 regions: primary motor, dorsolateral secondary motor, medial motor (SMA and pre-SMA), dorsolateral prefrontal cortex, language motor, primary somatosensory, parietal association, precuneus, primary and secondary auditory areas (auditory), inferior temporal cortex, object and face recognition areas, primary and secondary visual areas (visual), and limbic association areas (*Figure 5C*).

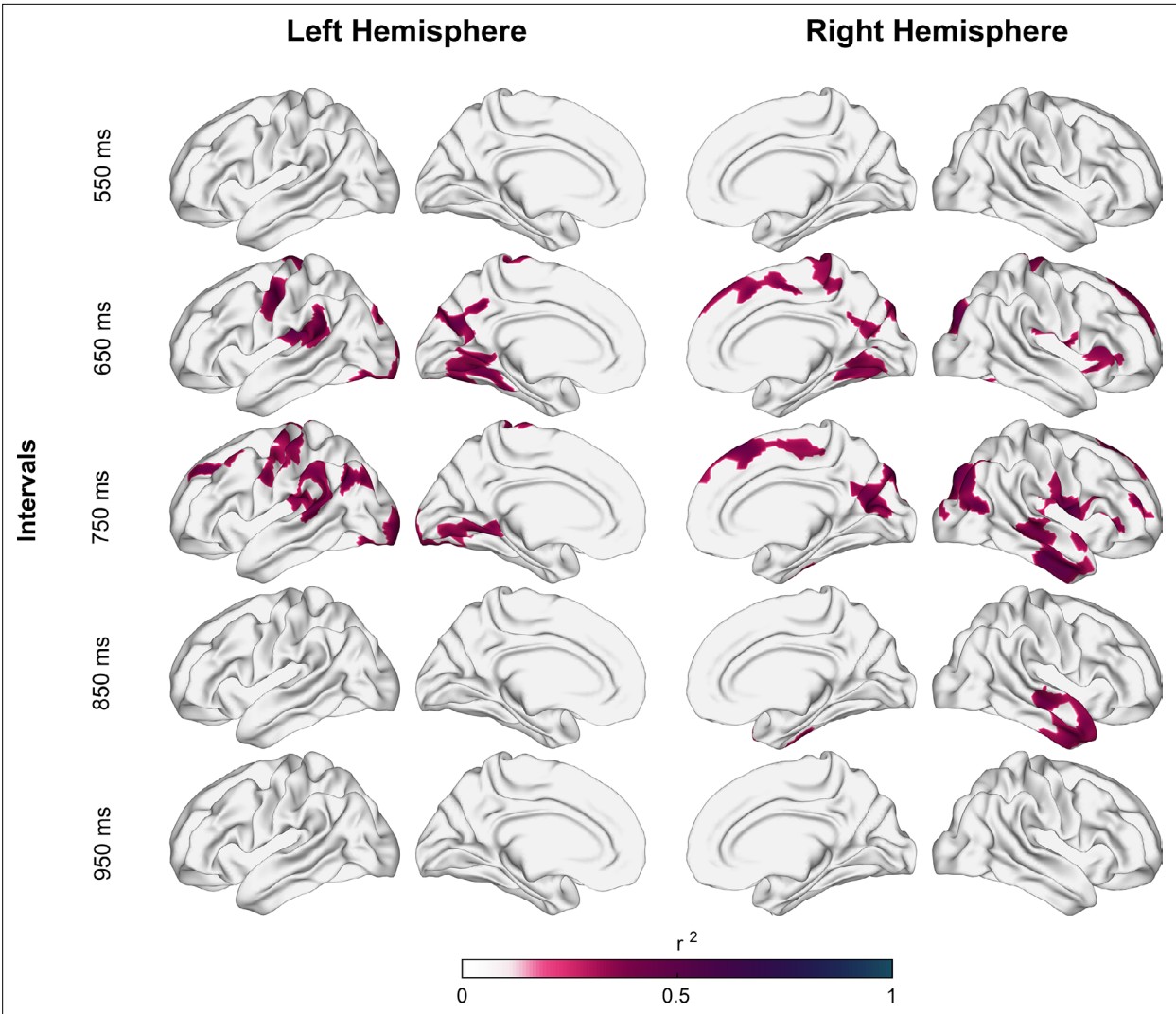

**Figure 4.** Coefficient of determination between auditory asynchronies and tangential apparent fiber density for each vertex in superficial white matter (sampled at 1.5 mm below the gray/white matter interface) across all tested intervals in the synchronization-continuation task (SCT) for both hemispheres. Significant correlations were localized in large clusters within motor, auditory, and visual areas, particularly for 650 and 750 ms intervals.

The online version of this article includes the following figure supplement(s) for figure 4:

**Figure supplement 1.** Association between asynchronies (650 ms intervals) and tangential superficial white matter apparent fiber density ($_{tan}$AFD) at various depths from the gray/white matter interface.

## Canonical correlation between behavioral and AFD maps

In the previous section, we correlated many behavioral measures with all vertices of the AFD maps, risking inflation of type I error. To address this, we performed a canonical correlation analysis (rCCA) between the behavioral data of the SCT and the structural information of the SWM (see Methods). This approach allowed us to independently assess the correlation between our AFD measurements of every vertex and every variable of the SCT. Specifically, rCCA was calculated between the matrix of behavioral parameters from the synchronization phase of the SCT (i.e., absolute asynchrony, constant error, temporal variability, and lag 1 autocorrelation) for each sensory modality (auditory and visual), every target interval (550–950 ms) and the AFD metrics from all vertices across the brain surface. Separate models were built for each AFD metric (i.e., $_{total}$AFD, $_{tan}$AFD, and $_{par}$AFD). Notably, all pairings of behavioral and AFD data rendered highly correlated canonical variates (*Figure 6A*). In line with the previous results, the highest correlation was found between the SCT data and the $_{tan}$AFD, closely followed by $_{total}$AFD, and then $_{par}$AFD (*Figure 6A*). The correlations between each behavioral

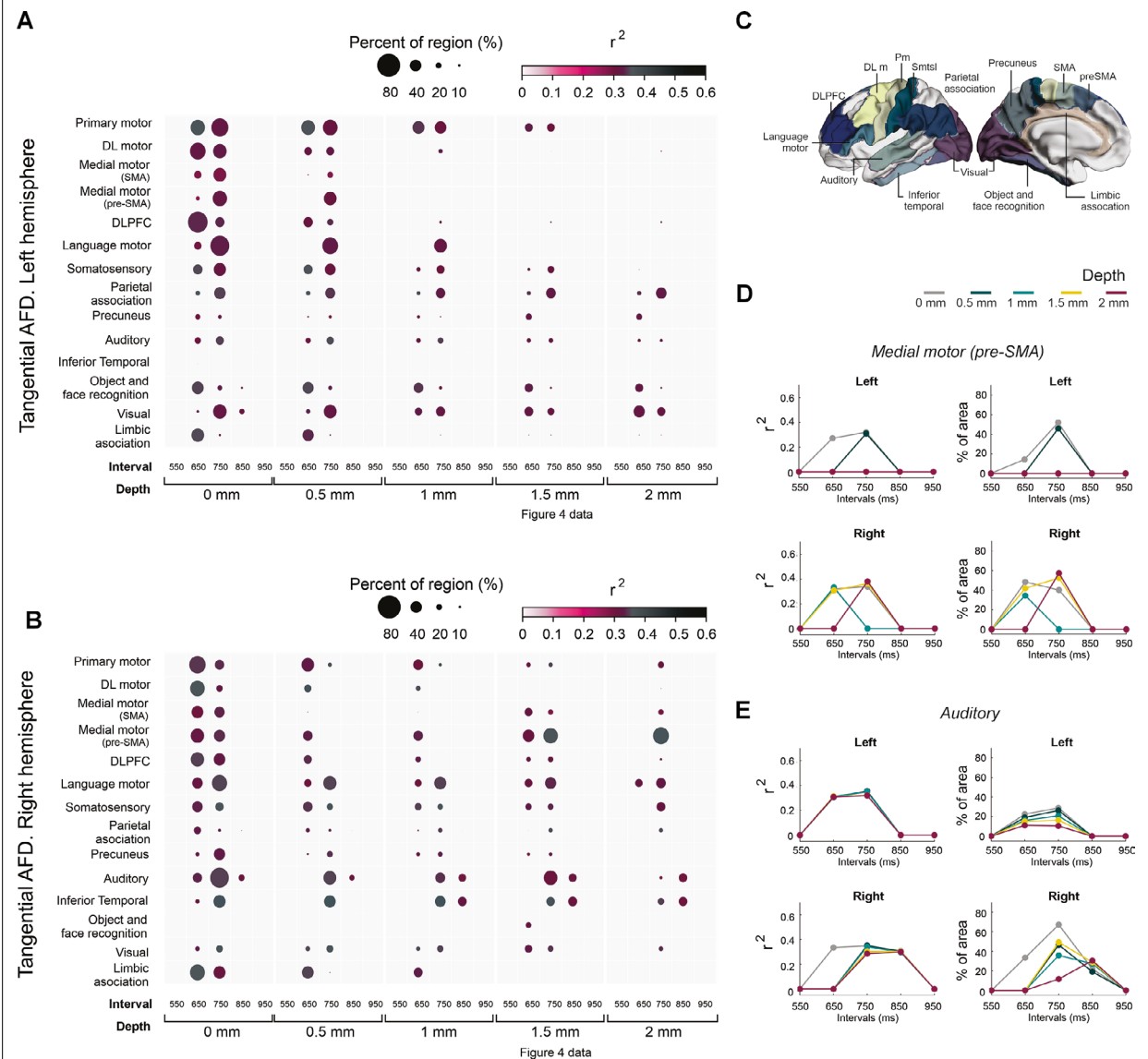

**Figure 5.** Associations between the asynchronies and regions of interest of $_{tan}$AFD. (**A, B**) Interregional correlation plot of the auditory asynchronies and $_{tan}$AFD as a function of the instructed interval and depth of the $_{tan}$AFD. Significant clusters were aggregated into fourteen regions (y-axis) based on the Brainnetome Atlas shown in C. The color and size of the circle for each cluster correspond to the correlation coefficient of determination (critical value $r > 0.355$ at $p < 0.02$, df = 29) and the percent of significant vertices in each area, respectively. The left hemisphere (**A**) showed more areas with significant vertices than the right (B; 17541>16641). A systematic decrease in circle size as a function of depth was observed across areas of both hemispheres. At a depth of 1.5 mm, the regions with a larger percent of significant vertices for the right hemisphere at 750 ms were the medial premotor (supplementary motor area [SMA] and preSMA), auditory, and language motor areas (**B**). In contrast, few significant vertices were observed across the regions of the left hemisphere (**A**). (**C**) Brainnetome Atlas (*Fan et al., 2016*) showing the 14 regions of interest (ROIs) analyzed in A and B. (**D**) Interval selectivity curves for the correlation coefficients and percent of significant vertices across $_{tan}$AFD depths (color coded) for preSMA. (**E**) Same as D but for the auditory cortex. Note that the preferred interval in the two areas is between 650 and 750 ms.

The online version of this article includes the following figure supplement(s) for figure 5:

**Figure supplement 1.** Interregional correlation plot of the auditory asynchronies and $_{par}$AFD as a function of the instructed interval and the depth of the $_{par}$AFD calculation.

parameter and $_{tan}$AFD and their corresponding canonical variates revealed a clear structure between the predictive behavior of subjects and the structural differences in U-fibers of the white matter in the audiomotor system. *Figure 6B* depicts the correlations between each standardized SCT parameter and the corresponding canonical variate (U), where it is evident that the asynchronies of the auditory

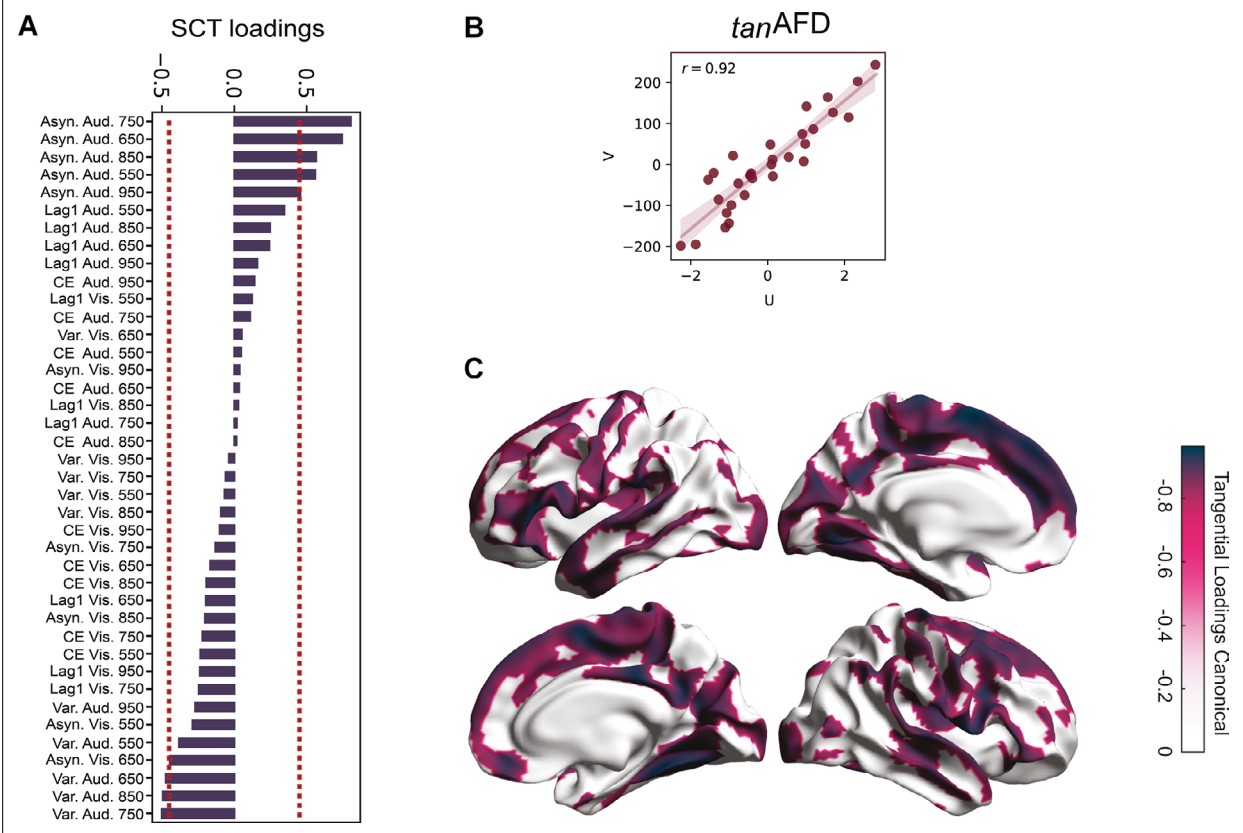

**Figure 6.** Canonical correlation between behavioral metrics and $_{tan}$AFD. (**A**) Loadings (correlations) of the synchronization-continuation task (SCT) variables for the $_{tan}$AFD model. Note the absolute asynchronies of the auditory modality showed the highest correlations with the structural data, at the 650–850 ms intervals, in line with previous results. In addition, this approach identified the total variability in the same sensory modality and intervals as significant, although with a lower correlation coefficient and of the opposite sign. (**B**) Pearson's correlation of canonical variates (U = behavioral; V = $_{tan}$AFD across vertices).(**C**) Loadings of the $_{tan}$AFD map. Note that the audiomotor system is highly correlated with the SCT behavior.

modality for the 650–850 ms intervals are the parameters with a significant relation to the canonical variate. A novel result from the rCCA is the significant association between the temporal variability of the auditory phase in the same intervals (650–850 ms). In addition, the $_{tan}$AFD map shows a significant correlation between the $_{tan}$AFD in audiomotor structures and the canonical variate. Again, the correlation between all vertices and their canonical variates was negative, corroborating the hypothesis that subjects with greater predictive abilities had a larger $_{tan}$AFD in the audiomotor circuit (*Figure 6C*).

## DWM and structural selectivity to the interval

We also evaluated the association between the precision and accuracy of the SCT tapping period and phase and the DWM properties. Fixel-based analysis (FBA) (*Dhollander et al., 2021*; *Raffelt et al., 2017*) was used to estimate micro- and macrostructural differences within DWM voxels (*Genc et al., 2018*; *Kelly et al., 2020*; *Mito et al., 2018*; *Rau et al., 2019*). This method, similar to that of our SWM analysis, is based on the CSD of DWI data.

FBA provides three fiber-specific indices (fiber density, fiber cross-section, and fiber density and cross-section; FD, FC, and FDC, respectively) (*Raffelt et al., 2017*). FD is derived from the integral of the fiber orientation distribution (FOD) lobes and is proportional to the total intra-axonal volume, thus reflecting the density of a population of fibers within a voxel (*Rojas-Vite et al., 2019*). Note that FD is identical to $_{total}$AFD used in our surface-based analyses, with nomenclature for FBA following *Raffelt et al., 2017*. If more than one fiber population coexists in a given voxel, the FOD is segmented, and an FD is assigned to each population, referred to as a fixel (fiber element). FC is a macroscopic metric of the fiber bundle diameter and, finally, FDC is a combination of FD and FC (see Methods). Briefly, the FBA analysis pipeline consists of five steps. First, the images of each subject are processed to obtain

a white matter FOD map in native space using CSD. Second, an FOD template and a fixel mask are built. Third, the AFD map and the corresponding fixels for each voxel are computed for each subject in native space. The FD metric is obtained for each fixel computed from the total DWI signal per voxel. Fourth, the fixels and AFD map are reoriented to the template. Finally, fixel-wise statistics are performed at each spatial location in template space.

The FBA revealed significant negative correlations between the FDC in the CC and the absolute asynchronies to the auditory metronome for 650, 750, 850, and 950 ms intervals (*Figure 7*). Therefore, this analysis showed a tight relationship between the density and bundle diameter of CC fibers and beat entrainment. Subjects exhibiting large phase accuracy with asynchronies close to zero also showed large FDC values, and subjects with poor phase accuracy and large asynchronies had low FDC values (*Figure 7—figure supplement 1*). As in the case of the U-fiber metrics, the FBA values were not correlated with period accuracy or precision, nor with the error correction for the auditory and visual conditions during the synchronization and continuation epochs.

The association between entrainment phase and white matter properties defined an interval-selective map in the CC, with the FDC at different levels of the CC showing significant correlations with the absolute asynchronies at specific intervals (*Figure 7*). This map showed an anterior–posterior gradient, with behavioral and structural associations for short and long intervals in the anterior and posterior portions of the CC, respectively. Thus, the FDC fixel values of the posterior midbody of the CC (interconnecting motor and premotor cortices and M1) showed a significant negative correlation with absolute auditory asynchronies for the 650 and 750 ms intervals (*Figure 7BC*; family-wise error-corrected p-value <0.05). For the asynchronies at the intermediate interval of 850 ms, a negative correlation was observed with FDC fixel values located in the isthmus and the splenium (*Figure 7C*; interconnecting primary motor, temporal, and visual cortices). Finally, the asynchronies of the 950 ms interval showed a significant negative correlation with fixels located in forceps minor and major (*Figure 7A, D*; interconnecting prefrontal and visual cortices, respectively).

It is evident in *Figure 7A–E* that the streamline segments of the fixels with entrainment correlations were located mainly at the joint of the two hemispheres across the CC. Nevertheless, a lateralization effect was found for the left hemisphere, with fixels associated with auditory asynchronies of 750 and 950 ms in the isthmus and splenium, respectively. In addition to the CC, the right fornix showed a significant association with the asynchronies of the 750 ms interval for the FD metric (see *Figure 7—figure supplement 1*).

Lastly, we carried out a correlation analysis between the mean absolute asynchronies across the five intervals and the FC (*Figure 8*) and FDC (see *Figure 7—figure supplement 1*). Notably, for the auditory condition, the tracts with significant FC fixels were the left arcuate fasciculus (*Figure 8A, E*), CC M1 (*Figure 8C, E*), forceps major (*Figure 8B, D*), superior longitudinal fasciculus 2 (*Figure 8F*), and right fornix (see *Figure 7—figure supplement 1*).

## Discussion

The present research determined the intersubject association between the structural properties of the SWM and DWM and different measures of rhythmic timing during a synchronization-continuation tapping task. Our study supports five conclusions: First, the tapping phase and period during the SCT showed precision and accuracy, as well as error corrections that were biased toward auditory rather than visual metronomes, thereby confirming previous observations. Second, the right audiomotor system exhibited individual differences in SWM and U-fiber density. These differences were correlated with the degree of phase accuracy of the tapping synchronization across subjects. Notably, the correlations were selective for the synchronization epoch of the auditory condition and were specific to the 650 and 750 ms intervals. Third, there was a significant association between the rhythmic entrainment phase and the density and bundle diameter of the CC, forming an interval-selective map with an anterior–posterior trend. This implies that the behavioral and structural associations for short and long intervals tended to be in the anterior and posterior portions of the CC, respectively. Fourth, the fiber bundle diameter of the arcuate fasciculus, CC, forceps major, and superior longitudinal fasciculus showed a significant correlation with the mean asynchronies across all tested tempos. Finally, we found no significant associations between SWM and DWM properties and temporal variability, constant error, or lag 1 autocorrelation under the visual and auditory conditions during the synchronization and continuation epochs of the SCT. These last findings suggest that connectivity within the

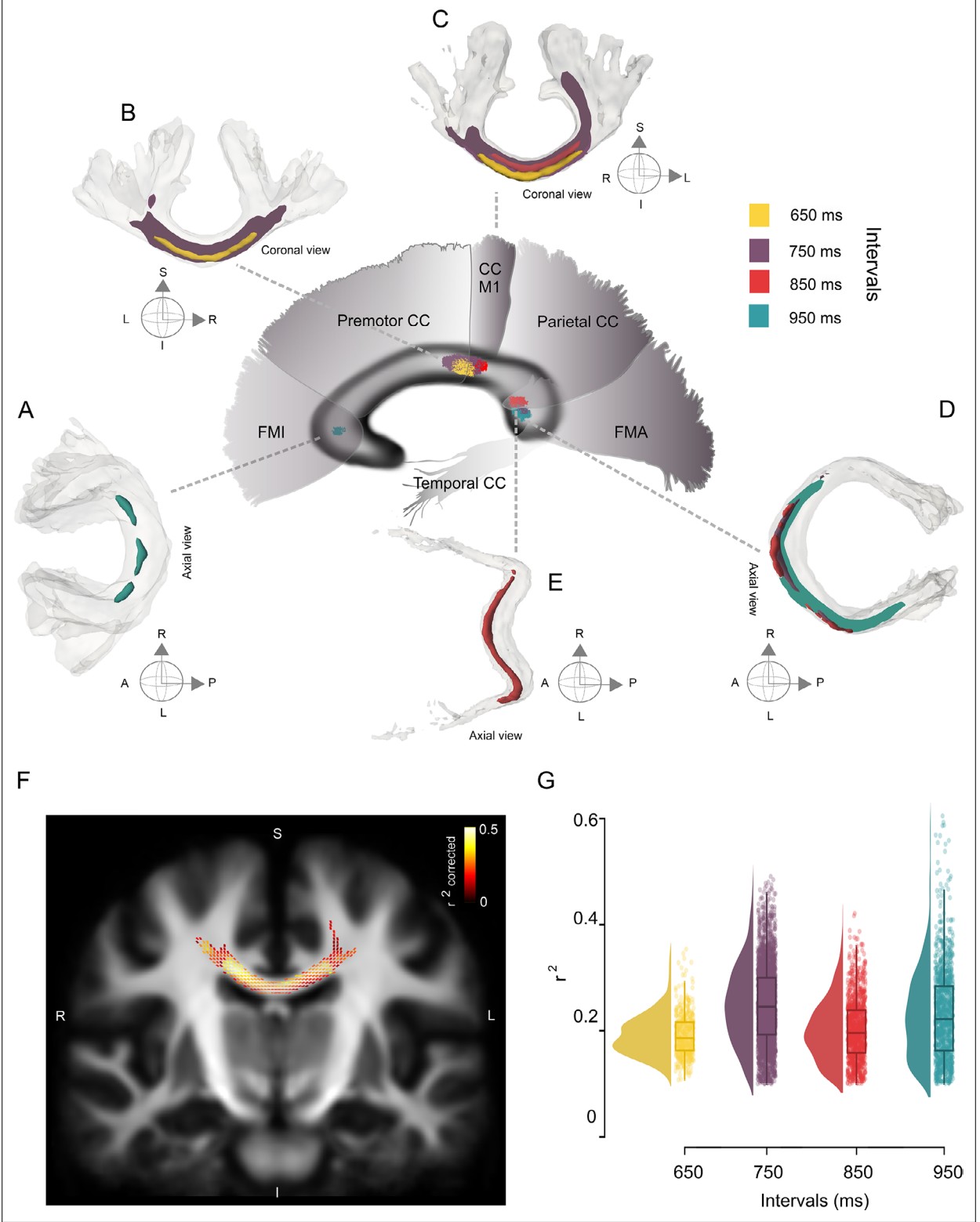

**Figure 7.** Interval-selective map of the correlations in the deep white matter. (**A–E**) Fiber bundles that showed significant correlations between the asynchronies in the synchronization-continuation task (SCT) auditory condition and the fiber density cross-section (FDC) of the corpus callosum (CC). Panels A–C correspond to the anterior coronal sections of the sagittal map depicted in the center of the figure. Panels D and E correspond to the posterior axial sections of the same central sagittal map. An interval-selective map with an anterior posterior gradient is depicted. (**F**) Coronal section of the CC showing the fixels with a significant correlation coefficient (color-coded *r* values; only fixels with pcorr < 0.05 are shown) between asynchronies at

*Figure 7 continued on next page*

*Figure 7 continued*

the 750 ms interval in the auditory condition and FDC. (**G**) Distribution of the coefficients of determination ($r^2$) of the FDC vs absolute asynchronies for the four intervals listed on the *x*-axis. The interquartile box plots are depicted on the right.

The online version of this article includes the following figure supplement(s) for figure 7:

**Figure supplement 1.** Spatial extent of deep white matter findings.

audiomotor system is tightly linked with the ability to synchronize in phase at the preferred tempo for auditory metronomes. We speculate that the structural white matter is associated with tapping-phase control, not with period representation, as the latter depends on neural population dynamics within the timing network (*Gámez et al., 2019*; *Betancourt et al., 2023*) and the former depends on a prior ability to detect changes in phase at the preferred tempo, whose intrinsic nature is defined in the audiomotor connectivity.

Many studies have shown that performance tapping synchronized to an auditory metronome is more precise and accurate than synchronization to a flashing visual metronome with the same timing characteristics (*Chen et al., 2002*; *Hove et al., 2013*; *Merchant et al., 2008c*; *Patel et al., 2005*; *Repp and Penel, 2004*; *Zarco et al., 2009*). This auditory–visual asymmetry can be cancelled out by visual moving metronomes (*Hove et al., 2010*; *Pérez et al., 2023*). Since the first processing relays, the auditory system has higher temporal resolution compared to the visual system (*Duysens et al., 1996*; *He et al., 1997*; *Sayegh et al., 2011*) and plays a critical role in time perception and reproduction across many tasks, not only in tapping SCTs (*Grondin et al., 2005*; *Merchant et al., 2008b*; *Merchant and de Lafuente, 2014c*). For example, when audiovisual stimuli are used in an oddball paradigm, the perceived duration is dominated by the auditory modality (*Chen and Yeh, 2009*). In addition, transcranial magnetic stimulation (TMS) disruption of the auditory cortex impaired time estimation for auditory and visual stimuli, while disruption in the visual cortex only produced timing impairments for visual stimuli (*Kanai et al., 2011*). Consequently, a current hypothesis is that the auditory cortex is engaged in multimodal temporal processing, and the interaction between the auditory and motor systems

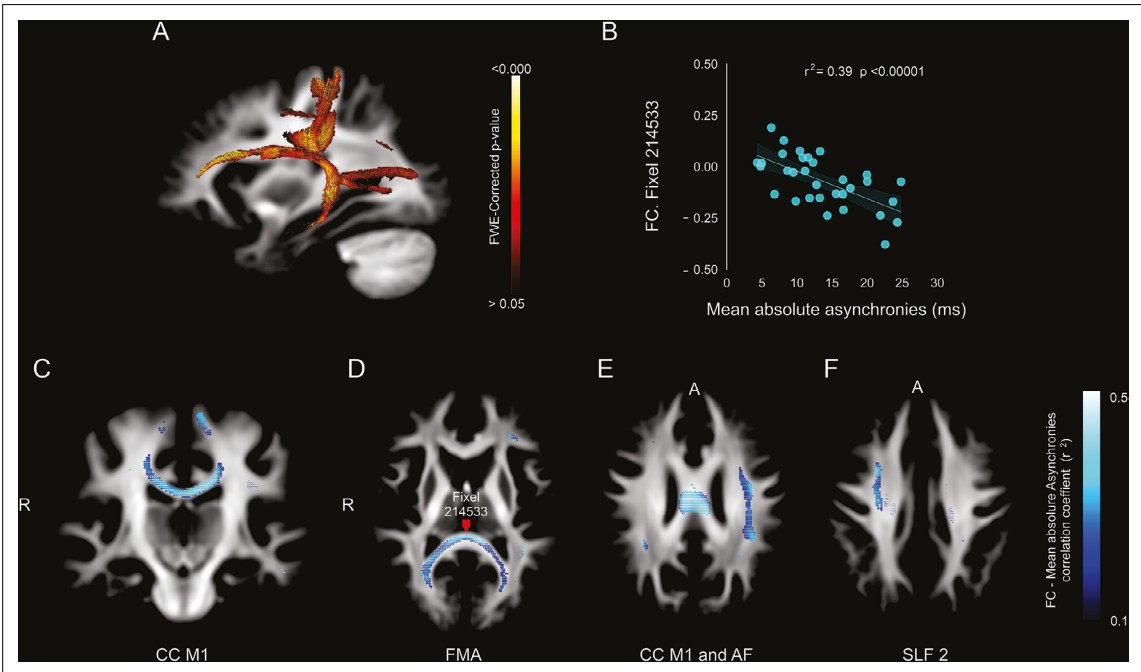

**Figure 8.** Association between deep white matter and mean absolute asynchronies across all target intervals. (**A**) Sagittal brain section showing the p-value of significant fixels. (**B**) Example of the significant correlation between the fiber cross-section (FC) metric and the mean absolute asynchronies in a single illustrative fixel (red dot in panel **D**). (**C–F**) Axial sections showing the negative correlation (*r*) between FC and mean absolute asynchronies for significant fixels. CC = corpus callosum (**C**), FMA = forceps major (**D**), AF = arcuate fasciculus (**E**), SLF 2 = superior longitudinal fasciculus 2 (**F**).

The online version of this article includes the following figure supplement(s) for figure 8:

**Figure supplement 1.** Spatial extent of deep white matter findings.

in the frontal lobe allows not only time encoding but also time prediction (*Merchant and Honing, 2014b*; *Merchant and Yarrow, 2016*; *Patel and Iversen, 2014*; *Schwartze et al., 2012*). Our present findings support this notion in different ways. First, the rhythmic timing performance of our subjects was biased toward the auditory condition across different behavioral measures. Second, the right audiomotor system showed widespread and significant correlations between the density of superficial U-fibers and the degree of sensorimotor phase accuracy across subjects. These anatomo-behavioral associations are selective to the intervals in the 650–750 ms range. Third, the fiber bundle diameter of the left arcuate fasciculus, a key tract connecting the parietotemporal auditory system with the frontal lobe, showed a significant correlation with the mean asynchronies across all tested tempos. Furthermore, the clear clustering of subjects as either good or bad synchronizers in a syllabic isochronous entrainment task correlates with both the difference in the activation magnitude in frontal areas and the changes in white matter pathways (i.e., left arcuate fasciculus) that connect the auditory system with the premotor cortical system (*Assaneo et al., 2019*). Hence, our results accentuate the audio-motor structural foundation for rhythmic entrainment (*Honing and Merchant, 2014*; *Miyata et al., 2022*). The lack of structural associations in the visual condition is probably due to the larger variability in rhythmic tapping for this modality, making it difficult to infer statistical correlations between our metrics. As described above, the null anatomo-behavioral associations for the visual condition could also be due to the slow sampling rate of visual periodic temporal attention, with a sampling interval that is longer than our tested durations (*Zalta et al., 2020*).

The observed associations between the measured SWM and DWM properties and the accuracy in the synchronization phase were negative. This result indicates that the white matter parameter values were greater in subjects with asynchronies close to zero and an accurate tapping phase with the metronome than in subjects with large gaps in time between the stimuli and taps. These results support the theory that intersubject differences in rhythmic entrainment phase depend on micro- and macrostructural white matter properties, which could have a genetic and/or learned substrate. From a genetics perspective, we could speculate that the existence of poor and superior synchronizers (*Blecher et al., 2016*) may depend on the FD of superficial U-fibers in the right audiomotor system, as well as on the density of deep tracts such as the CC and arcuate fasciculus. From a training perspective, these SWM and DWM bundles may develop larger density and myelination during intense musical practice, distinguishing the audiomotor tracts between musicians and non-musicians (*Palomar-García et al., 2020*; *Vaquero et al., 2018*; *Zatorre et al., 2007*).

The correlations between the density of tangential U-fibers in the right audiomotor circuit and the asynchronies for the auditory condition were interval-selective for the intermediate tested tempos. The observed interval specificity corroborates the existence of an spontaneous rhythmic tempo, already observed in many studies of rhythmic entrainment, and with values between 600 and 750 ms (*Drake et al., 2000a*; *Drake et al., 2000b*; *McAuley et al., 2006*: *Dalla Bella et al., 2017*). The biological intrinsic periodicity may depend on a biased distribution of preferred tempos toward 1.5 Hz in the interval-tuned neurons of the motor system (*Bartolo et al., 2014*; *Bartolo and Merchant, 2009*; *Pérez et al., 2023*; *Pérez and Merchant, 2018*). Interestingly, *Zalta et al., 2020* showed that humans were best at following auditory rhythms at a rate of ~1.4 Hz and that overt motor activity optimizes auditory periodic temporal attention at a similar rate. This rate is close to the intervals with structural associations in the present study. Consequently, we suggest that cortico-cortical connectivity within the audiomotor system is especially designed to support the internal representation of an auditory beat at the preferred tempo, providing larger phase-locking abilities with the metronome for spontaneous motor tempos (*Balasubramaniam et al., 2021*). On the other hand, *Zalta et al., 2020* also demonstrated that the best rate for visual rhythms was far slower, at ~0.7 Hz (*Zalta et al., 2020*). Therefore, it is possible that we did not get effects in the visual condition because our target interval of 950 ms was not slow enough for this modality.

Ultra-high-field (7T) functional imaging revealed that the medial premotor areas (SMA and pre-SMA) of the human brain possess neural circuits that are tuned to different durations, forming a topographical arrangement during a visual discrimination task. These chronomaps show units with enhanced responses for the preferred interval and suppressed activity for the non-preferred duration, and define a topographical gradient with short preferred intervals in the anterior portion and long preferred intervals in the posterior portion of the medial premotor areas (*Protopapa et al., 2019*; *Schwartze et al., 2012*). *Protopapa et al., 2019* also showed chronomaps in the SMA during an

interval reproduction task using auditory cues, which is similar to the task used in this study. Notably, chronomaps also showed an anterior–posterior gradient, but they represent relative rather than absolute time, and they presented some flexibility in the location of the preferred interval depending on the task context (*Bueti et al., 2021*). In addition, a recent imaging study described the existence of large chronomaps covering the cortical mantle from the dorsal and ventral premotor areas to the occipital pole (*Harvey et al., 2020*; *Hendrikx et al., 2022*). Our measurements of the FDA also revealed a topographical arrangement in the correlations between FD in the DWM and CC and the sensorimotor phase accuracy of subjects. We also found significant anatomo-behavioral associations in the anterior part of the CC for short intervals and in the posterior CC for long tapping tempos (*Schwartze et al., 2012*). Nevertheless, our interval-selective map is defined by the correlation between asynchronies and FDA, with no topographic model of the distribution of preferred intervals as shown in functional magnetic resonance imaging (fMRI) studies. These studies have not explored individual differences. Moreover, we found a frontal CC cluster of fixels with longer interval selectivity, producing a discontinuity in the anterior–posterior gradient of preferred intervals. These findings suggest that the map for duration selectivity starts anteriorly in the CC, which is linked to the premotor system, and ends in the CC of the visual areas of the occipital lobe. The anterior selectivity for the 650 and 750 ms ISI supports the notion that the motor system functions at 1.7 Hz during beat perception and entrainment and strongly influences the auditory system but not the visual system at this tempo (*Zalta et al., 2020*). The latter could explain why we did not find interval selectivity in the CC for the visual condition. Consequently, timing maps define a cortical processing framework for efficient timing integration that has both functional and anatomical bases, especially for the auditory modality (*Merchant and de Lafuente, 2024a*).

Previous studies have shown correlations between sensorimotor synchronization task performance and the microstructural characteristics of DWM. For instance, *Blecher et al., 2016* found a positive association between the fractional anisotropy of the left arcuate fasciculus and CC and performance in an auditory-cued finger-tapping task. The SWM immediately below the cortex has received has been less studied than DWM fasciculi, even though it accounts for 60% of the total white matter volume and is pivotal in maintaining cortico-cortical connectivity (*Schüz and Braitenberg, 2002*; *Schmahmann et al., 2006*). Nonetheless, recent studies have implicated SWM abnormalities in epilepsy (*Liu et al., 2016*), autism spectrum disorder (*Hong et al., 2019*), Alzheimer's disease (*Phillips et al., 2016*), schizophrenia (*Nazeri et al., 2013*), and stroke (*Stockert et al., 2021*). The SWM contains short-range association fibers that connect adjacent gyri (U-fibers) and the initial or final portions of long-range connections that traverse the DWM (*Schüz and Braitenberg, 2002*; *Guevara et al., 2020*; *Kirilina et al., 2020*; *Yoshino et al., 2020*). The SWM is difficult to study because of its complicated geometry and abundance of fiber crossings (*Guevara et al., 2020*). To better characterize the SWM microstructure, we separated the two components of the SWM and performed a surface- and depth-wise evaluation. Complementarily, we analyzed the DWM using FBA, which addresses many of the shortcomings of voxel-wise analysis of diffusion tensor imaging (*Dhollander et al., 2021*). This two-pronged approach allowed us to evaluate the entire white matter volume and show the association between its mesoscopic characteristics (i.e., AFD) and predictive tapping synchronization.

A potential limitation of our study is the relatively small number of participants, related to the time-consuming nature of the behavioral evaluation and the long scanning time. Moreover, many statistical tests were performed, relating several behavioral metrics to various diffusion metrics across the brain at different depths. To minimize the possibility of statistical errors, we performed an rCCA to jointly model the behavioral and imaging metrics, thus accounting for the numerous statistical tests and reducing the possibility of reporting false positive findings. This reinforced the importance of superficial cortico-cortical communication through U-fibers and the SCT for intervals around the preferred tempo. While we have attempted to control for statistical errors as much as possible, a sample of 32 young adults with specific inclusion and exclusion criteria may inevitably not represent the population. We expect our current findings to be replicated and extended in future studies.

In conclusion, our results showed that the subjects' accuracy in SCT performance was associated with higher FC, FDC, or FD values. This is consistent with the literature that shows that better performance in different tasks is associated with higher values of FBA metrics. For example, better performance in a bimanual coordination task was associated with higher FBA values in the CC (*Zivari Adab et al., 2020*).

## Methods

This study was approved by the Ethics Committee of our Institution (049H-RM).

### Participants

Thirty-two healthy human subjects (age = 25.37 ± 3.21 years; 19 females) without musical training volunteered to participate and gave informed consent, which complied with the Declaration of Helsinki and was approved by our Institutional Review Board. All participants were right-handed and native Spanish speakers. They did not have MRI contraindications or neurological, psychiatric, or cognitive conditions.

### Apparatus

Participants were seated comfortably in a quiet experimental room, facing a high-definition 23″ touch screen (refresh rate: 60 Hz; ELO Touch solutions) located 50 cm away, which they were instructed to tap using the right index finger. Auditory stimuli were presented through noise-canceling headphones (Sony, MDR-NC50).

### Experimental task

The SCT of the present study is similar to the standard SCT described elsewhere (*Merchant et al., 2008b*; *Merchant et al., 2008b*). However, instead of tapping a button, the subject tapped on the right or left halves of the touch screen. The task started when three empty white circles (radius 1 degree of visual angle) on a black background were presented simultaneously, forming an inverted isosceles triangle (2 degrees of visual angle on each side). Subjects were trained to place their finger at the central-bottom target to start the trial and attend to a sequence of two to six right/left alternating stimuli with a constant interstimulus interval (perception epoch). They were instructed to tap the touchscreen on the corresponding target in synchrony with a metronome for six intervals (synchronization epoch) and continue tapping on the screen for another six intervals without the metronome (continuation epoch; *Figure 1*). During the perception epoch, the subjects were instructed to start tapping once they got the beat from the metronome, and they had two to six stimuli to start tapping. The mean produced interval was presented at the end of the trial as feedback. A trial was considered correct if, for every interval, the absolute difference between the produced interval and the target interval was below 30% of the target interval during the synchronization and below 40% during the continuation epoch. The task was programmed using Psychtoolbox for Matlab (2018, Mathworks).

### Stimuli

The visual metronome (33 ms, yellow circle, 0.5 degrees of visual angle) was alternatively displayed inside the right or left empty circles. The auditory metronome (33 ms , 550 Hz, 80–85 dB SPL) was also presented alternatively in the right or left headphone. The isochronous target intervals were 550, 650, 750, 850, and 950 ms, which were pseudorandomly presented within a block. Each subject performed 10 correct trials for each target interval.

### Procedure

The MRIs were collected in the first session. In a second visit, the auditory and visual tasks were performed randomly across subjects outside of the MRI scanner.

### Analysis of behavioral data

Four metrics were calculated to assess the subjects' performance during SCT (*Figure 2A*; *Gámez et al., 2018*; *Merchant et al., 2008c*). During the synchronization epoch, we estimated the absolute asynchronies and autocorrelations of the inter-tap interval time series (*Iversen et al., 2015*; *Wing, 2002*). The constant error and temporal variability were calculated from the produced intervals during both the synchronization and continuation epochs. Absolute asynchronies were defined as the unsigned difference between stimulus onset and tap onset. Constant error was calculated as the difference between the produced interval and the target interval and is a measure of timing accuracy. Temporal variability was defined as the standard deviation of the total produced intervals and is a metric of timing precision. The autocorrelation of the six inter-tap intervals during a trial was calculated and averaged across trials. Thus, lag 1 autocorrelation is normally negative for isochronous metronomes,

meaning that a longer produced interval tends to be followed by a shorter interval and vice versa, reflecting an error correction mechanism used to maintain the beat of the metronome during synchronization (*Iversen et al., 2015*). A repeated-measures ANOVA with two and three factors was carried out for the analysis of asynchronies, constant error, and temporal variability.

## Imaging protocol

Images were acquired using a 3T Philips Achieva TX scanner with a 32-channel head coil. T1-weighted volumes were obtained using a three-dimensionally encoded spoiled gradient echo sequence (repetition/echo times (TR/TE) = 8.2/3.7 ms, flip angle = 8°, field of view = $256 \times 256 \times 176$ mm$^3$, matrix size = $256 \times 240 \times 176$ mm$^3$ yielding voxel resolution = $1 \times 1 \times 1$ mm$^3$). DWIs were acquired with echo-planar imaging, $2 \times 2 \times 2$ mm$^3$ voxel resolution, FOV = $256 \times 256$ mm$^3$, 62 axial slices, TR/TE = 16,500/72 ms. Images were sensitized to diffusion with $b = 1000$ s/mm² (64 unique directions) and $b = 3000$ s/mm² (96 directions). Five volumes without diffusion weighting ($b = 0$ s/mm²) were also acquired, along with an additional $b = 0$ s/mm² volume obtained with reversed-phase encoding polarity for correction of geometric distortions.

## Image processing

a.  Cortical surface. T1-weighted images were used to estimate cortical surfaces. Images were first denoised (*Coupe et al., 2008*) and corrected for intensity inhomogeneities (*Tustison et al., 2011*). Binary masks of the brain were obtained with volBRAIN v.2.0 (*Manjón and Coupé, 2016*). Cortical surfaces were obtained through the FreeSurfer pipeline v.6.0 (*Fischl, 2012*). Individual surfaces were registered to the surface template with 20,484 vertices (fsaverage5).
b.  DWIs were first denoised (*Manjón et al., 2013*) and bias field corrected (*Tustison et al., 2011*), then corrected for geometric distortions and motion using FSL's topup-eddy algorithm (*Andersson and Sotiropoulos, 2016*).

## Fixel-based analysis

We analyzed individual fiber-specific properties in the presence of crossing fiber populations ('fixels'; *Raffelt et al., 2015*) following the steps described in *Raffelt et al., 2017* and using the tools available in MRtrix3 (*Tournier et al., 2019*). A white matter mask was computed for each subject, followed by global signal intensity normalization of the DWI, which was performed across subjects by dividing all volumes by the median $b = 0$ s/mm² intensity. Images were upsampled to 1 mm$^3$ isometric resolution (*Dyrby et al., 2014*). White matter FODs were estimated using the multi-shell, multi-tissue CSD (MSMT-CSD) algorithm (*Jeurissen et al., 2014*). Tissue-specific response functions were calculated for each subject, from which we derived group-averaged response functions that were used to estimate FODs (lmax = 8) for each subject. An FOD template was constructed through iterative non-linear registration using the FODs of all 32 subjects followed by the calculation of the intersection of masks of all subjects. Fixels were derived at each voxel by FOD segmentation and reoriented to the study template. Finally, FBA metrics (FD, FC, and FDC) were calculated for each fixel.

## Statistical analysis

A whole-brain probabilistic tractogram was calculated based on the FOD population template, seeded from a whole-brain white matter mask to produce a tractogram of 20 million streamlines. Next, the SIFT algorithm (*Smith et al., 2013*) was used to select a subset of streamlines ($n = 2$ million) that best fit the diffusion signal and therefore reduce tractography biases. The structural connectivity metric between fixels was obtained according to probabilistic tractography using the connectivity-based fixel enhancement tool (*Raffelt et al., 2015*).

The FD, FC, and FDC measures were correlated with SCT for both conditions (visual and auditory) using a general linear model. Non-parametric permutation tests and connectivity-based fixel enhancement (*Raffelt et al., 2015*) were carried out for correction of multiple comparisons (*Nichols and Holmes, 2002*).

After the statistical analysis, tracts with significant fixels in the group space were identified using the tract-selection regions from the XTRACT tool (*Warrington et al., 2020*) included in FSL software (FMRIB's Software Library – FSL, Oxford, UK) (*Smith et al., 2004*), which were warped into our population template. Tracts with significant fixels were reconstructed using MRTrix (*Tournier et al., 2019*).

## SWM surfaces

To assess SWM properties, we used a synthetic representation of axons based on cortical topology. We computed a Laplacian potential field between the ventricles and the gray/white matter boundary (*Jones et al., 2000*; *Lerch et al., 2008*; *Liu et al., 2016*) using *minclaplace* (*Lerch et al., 2008*). Next, Laplacian streamlines were seeded at each vertex of the white matter surface and propagated toward the ventricles using Matlab, 2020A. Thus, one Laplacian streamline was obtained for every white matter surface vertex (*Figure 3A*). The distance between each step of the Laplacian streamline was 100 μm, and streamlines were truncated at 5 mm. This resulted in smooth and non-overlapping pathways that respect topology and traverse the SWM. Furthermore, the first segment of these streamlines is perpendicular to the gray-white matter (GM–WM) surface, with subsequent segments gradually bending as they extend away from it. These aspects make the Laplacian streamlines behave similarly to what is expected from the anatomy of the SWM. Finally, all the white matter surfaces and synthetic streamlines were warped to their corresponding subject-specific DWI space via between-modality non-linear registration using ANTS (*Avants et al., 2011*). Data for one subject was discarded for SWM analysis (*Figures 3–5*) due to suboptimal registration between DWI metric maps and white matter surface.

We used the fixel-based information to independently evaluate the two fiber systems that coexist in the SWM: U-fibers subserving short-range cortico-cortical connectivity and long-range projection, association and commissural fibers (*Kirilina et al., 2020*; *Schüz and Braitenberg, 2002*; *Yoshino et al., 2020*). With the assumption that U-fibers run tangentially to the gray/white matter surface and long-range fibers impinge on the surface in a perpendicular fashion, we attributed the fixels oriented parallel to the segments of the Laplacian streamlines to long-range fibers, and the remaining fixels to the U-fiber system. DWI-derived metrics were sampled along each Laplacian streamline at 0, 0.5, 1, 1.5, and 2 mm under the gray/white matter interface. The metrics sampled were $_{total}$AFD (the integral of all FODs within a given voxel) and AFD attributed to either long-range fibers ($_{par}$AFD; the integral of the FOD of the fixel parallel to the Laplacian streamline) or U-fibers ($_{tan}$AFD; defined as $_{total}$AFD − $_{par}$AFD).

All metrics along the Laplacian streamlines were projected onto the gray/white matter surface of the fsaverage5 template for visualization and statistical analyses and smoothed using a two-dimensional kernel of 15 mm of full width at half maximum.

## Surface-based analysis of SWM

Analyses were performed by fitting a general linear model at each vertex using SurfStat (https://www.math.mcgill.ca/keith/surfstat). This analysis assessed the relation between the value of diffusion metrics in each vertex (*i*) and the behavioral metrics from the SCT (absolute asynchronies, constant error, temporal variability, and lag 1 of the autocorrelation of the inter-tap-interval time series), as:

$$\text{SWM}_i = \beta_0 + \beta_1^* \text{STC}_{\text{metric}}$$

Surface vertex-wise analysis was corrected for multiple comparisons using random-field theory with a cluster-forming threshold pcft < 0.001 (*Eklund et al., 2016*). Clusters with $p_{cluster}$ < 0.001 were deemed significant (https://www.math.mcgill.ca/keith/surfstat).

Significant clusters were anatomically identified using the Brain Atlas Based on Connectional Architecture (Brainnetome) (*Fan et al., 2016*). All the analyses were carried out in Surfstat (*Worsley et al., 2009*) for Matlab (2018; Mathworks).

## Regularized canonical correlation analysis

As an additional verification of the results obtained via random-field theory analysis, we performed a canonical correlation analysis between the behavioral data of the SCT and the structural information of the SWM. This approach allowed us to independently assess the correlation of the AFD measurements of every vertex with every variable of the SCT. Concretely, rCCA was calculated between the matrix of behavioral metrics from the synchronization phase of the SCT (i.e., absolute asynchrony, constant error, temporal variability, and lag 1 autocorrelation) for each sensory modality (auditory and visual), and every target interval (550–950 ms) and the AFD matrix of the whole brain. Given the orthogonality between $_{tan}$AFD and $_{par}$AFD, and their collinearity with $_{total}$AFD individual models were built for every AFD metric.

Since the number of variables (particularly the number of vertices) is much larger than the sample size, we included two regularizing parameters for the covariance matrices in the model (rCCA) (*Mihalik et al., 2022*). These parameters were optimized by a grid search algorithm that maximized the correlation of the canonical variates (*Figure 6A*). Confidence intervals (at 99%) for the loadings of the variables after model fitting were estimated by building null distributions of loadings based on random permutations ($n$ = 10,000) of the original metrics. This was useful to identify the most relevant behavioral variables (*Figure 6*) associated with the structural data.

## Acknowledgements

We thank Sonja Kotz and Florencia Assaneo for their valuable comments on the manuscript and to Jessica Gonzalez Norris for proofreading the manuscript. We also thank Luis Prado, Erick Pasaye, Juan Ortiz, and Leopoldo González for their technical assistance.

## Additional information

### Competing interests

Hugo Merchant: Reviewing editor, *eLife*. The other authors declare that no competing interests exist.

### Funding

| Funder | Grant reference number | Author |
| --- | --- | --- |
| Consejo Nacional de Humanidades, Ciencias y Tecnologías | A1-S-8330 | Hugo Merchant |
| Dirección General de Asuntos del Personal Académico, Universidad Nacional Autónoma de México | PAPIIT IG200424 | Hugo Merchant |
| Secretaria de Ciencia y Tecnología, Ciudad de México | 2342 | Hugo Merchant |
| Consejo Nacional de Humanidades, Ciencias y Tecnologías | C1782 | Luis Concha |
| Consejo Nacional de Humanidades, Ciencias y Tecnologías | FC-218-2023 | Luis Concha |
| Dirección General de Asuntos del Personal Académico, Universidad Nacional Autónoma de México | PAPIIT AG200117 | Luis Concha |
| Dirección General de Asuntos del Personal Académico, Universidad Nacional Autónoma de México | IN213423 | Luis Concha |
| Consejo Nacional de Humanidades, Ciencias y Tecnologías | 280464 | Pamela Garcia-Saldivar |

The funders had no role in study design, data collection, and interpretation, or the decision to submit the work for publication.

## Author contributions
Pamela Garcia-Saldivar, Conceptualization, Data curation, Software, Formal analysis, Investigation, Visualization, Methodology, Writing - original draft, Writing – review and editing; Cynthia de León, Software, Investigation, Methodology, Writing – review and editing; Felipe A Mendez Salcido, Formal analysis, Methodology; Luis Concha, Software, Formal analysis, Supervision, Validation, Investigation, Visualization, Methodology, Writing – review and editing; Hugo Merchant, Conceptualization, Formal analysis, Supervision, Funding acquisition, Validation, Investigation, Writing - original draft, Writing – review and editing

## Author ORCIDs
Pamela Garcia-Saldivar ⓘ https://orcid.org/0000-0003-3274-4955
Cynthia de León ⓘ https://orcid.org/0000-0002-4488-2864
Felipe A Mendez Salcido ⓘ http://orcid.org/0000-0002-1697-5203
Luis Concha ⓘ http://orcid.org/0000-0002-7842-3869
Hugo Merchant ⓘ https://orcid.org/0000-0002-3488-9501

## Ethics
Thirty-two healthy human subjects without musical training volunteered to participate and gave informed consent, which complied with the Declaration of Helsinki and was approved by our Institutional Review Board. This study was approved by the Ethics Committee of the Institute of Neurobiology, Universidad Nacional Autónoma; noma de México, Campus Juriquilla with the number 049H-RM.

## Decision letter and Author response
Decision letter https://doi.org/10.7554/eLife.83838.sa1
Author response https://doi.org/10.7554/eLife.83838.sa2

---

# Additional files

## Supplementary files
• MDAR checklist

## Data availability
Data is available at OSF: https://doi.org/10.17605/OSF.IO/YNVF3.

The following dataset was generated:

| Author(s) | Year | Dataset title | Dataset URL | Database and Identifier |
| --- | --- | --- | --- | --- |
| Garcia-Saldivar P, Merchant H, Concha P | 2024 | MRI Data. White matter structural bases for phase accuracy during tapping synchronization | https://doi.org/10.17605/OSF.IO/YNVF3 | Open Science Framework, 10.17605/OSF.IO/YNVF3 |

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
