## [Editor Report]

This paper is valuable in that it provides a critical missing link between measures of structural connectivity and rhythmic tapping abilities, pointing to interesting possibilities for how tapping synchronization is carried out. The methodology and findings are convincing, and of interest to those studying the neural mechanisms of timing.

---

## [Decision Letter]

**Decision letter after peer review:**

Thank you for submitting your article "White matter structural bases for predictive tapping synchronization" for consideration by *eLife*. Your article has been reviewed by 2 peer reviewers, and the evaluation has been overseen by a Reviewing Editor and Barbara Shinn–Cunningham as the Senior Editor. The reviewers have opted to remain anonymous.

Essential revisions (for the authors):

1) The statistical approach will need streamlining and adequate correction for respective families of hypotheses where appropriate. As indicated by the reviews, Having multiple outcome measures also poses a potential risk of inflated type I error. See esp. comments by Rev #1 and Rev #2 in this regard! Sample size justifications and corresponding tempering statements/quantifications about statistical power are probably in order as well.

2) More background information is needed for the structural MR data and analyses, see esp. Rev # 2.

3) The reviewers' repeated calls for overall editing for clarity, in terms of structure as well as grammar, is not to be taken lightly. Thank you.

*Reviewer #1 (Recommendations for the authors):*

– By far, one of the most major things to note here is that there are a wide variety of statistical tests carried out across levels. That is, there are 4 behavioral measures – not to mention multiple intervals and modalities and phases, 3 fiber density measures, at 5 depths, as well as other DWI measures tested. Was there any attempt to control for these multiple comparisons? I note that this is different from the within–brain multiple comparisons (i.e. across voxels).

– The second major claim of the paper is the chronotopic organization in the corpus callosum. I find it difficult to assess whether this is true or not. From my observation, there are certainly different parts associated with different intervals, but it does not seem to go in a gradient direction (anterior–posterior, as the authors state). For example, the 750ms interval appears to cover most of the area, and the 950ms interval seems to be in a completely separate area. It's possible that there is a gradient here, but the authors have not convincingly presented it; perhaps by collapsing across dimensions this could be shown, but otherwise, any claims of "chronotopy" similar to what Protopapa and colleagues observed should be tempered.

– There is, of course, concern about the sample size in this study. A group of 32 subjects can certainly show associations, but only if they are quite high and potentially subject to bias. This is not meant to say that the results are certainly false, but merely that there is an exploratory aspect to the findings that should be emphasized.

– The motivation is a bit awkward and amounts to: "we don't know the role of structural connectivity in rhythmic entrainment; we hypothesize that structural connectivity accounts for individual differences in rhythmic entrainment". That is, we don't know if X is involved, therefore we hypothesize that X is involved? Why? While it's true that little work has been done in this specific area, the hypothesis is quite unformed.

– The authors should likely read and cite a paper by Zalta, Petkoski, and Morillon (2018) on periodic temporal attention. Here, the authors observed that humans were best at attending to auditory rhythms at a rate of ~1.4Hz, very close to the rate observed in the present study. Moreover, for visual rhythms, the best rate was far slower, at ~0.7Hz. It is possible then that the reason for the lack of effects in the visual condition is that the authors did not test any intervals that were slow enough.

Zalta, A., Petkoski, S., and Morillon, B. (2020). Natural rhythms of periodic temporal attention. Nature communications, 11(1), 1–12.

– How was the feedback conveyed? The methods state they are the mean asynchrony, but was this an offset value? A proportion? Did subjects know if they were tapping too fast or too slow?

– There is some concern regarding the latencies for which the tapping information is collected. If the monitor was running at 60Hz, is this the same sample rate for the taps? In this case, there might be a concern that the measures being collected are not sufficiently precise to reveal individual differences, which may explain why there were so few correlations except for asynchrony.

– What if subjects tapped the monitor but missed the target? Was the tap not collected? Did all subjects remain within the target?

– More detail about the FWE correction is needed. Was this at the cluster level? Voxel level? The text describes "FWE <0.0.001" and "Cluster<0.001", so what does this mean, precisely? Also, the methods state at one point that a "permutation test" was done, then at another point state random field theory

– For the continuation phase of the task, did the authors consider decomposing the inter–tap–interval variance using the Wing and Kristofferson (1973) model? This would provide measures of putative "central" and "motor" variance. Given that multiple intervals were tested, the authors should observe that central variance grows with the interval, whereas motor variance remains flat. Further, one might expect motor variance to relate more to connections within motor regions, whereas central variance could relate to audiomotor connections.

– For continuation performance, a commonly observed phenomenon is "drift" in which subjects continuously speed up or slow down their responses. Was this observed at all? A typical method for drift correction is to detrend the time series with a first–order polynomial and use the residuals for calculating measures of variance (Vorberg and Wing, 1996). Was this examined at all?

Vorberg, D., and Wing, A. (1996). Modeling variability and dependence in timing. In Handbook of perception and action (Vol. 2, pp. 181–262). Academic Press.

– Another prediction of the W–K model is the negative lag 1 autocorrelation which the authors examine here. Yet, violations of this commonly do occur. Were there violations (i.e. trials) where subjects did not show negative autocorrelations?

*Reviewer #2 (Recommendations for the authors):*

Interesting study, a nice simple task, and interesting white–matter metrics that I do not see often in this literature.

The selection of interval ranges needs further justification–why did the researchers choose these particular interval lengths, and did they predict differences between them? The preferred tempo range includes the 650 ms interval, as they note, but most authors would suggest that the 550 ms interval rate would be in this range for many people as well, as preferred beat rates tend to span 300–900 ms (Parncutt, 1994). Thus, was the range of intervals selected to provide evidence of structural chronomaps in the brain, or did the researchers expect to see behavioural differences across the intervals (that may be related to structural brain measures) that they felt would be meaningful (being more or less related to optimal tapping rates), and therefore they used a range of intervals to assess this?

There are 2 modalities and 5 different interval lengths, all of which appear to be analyzed independently in the brain data. The MRI methods contain very little information about multiple comparisons correction related to the number of different tests that are performed–this needs much clearer explication.

Unsigned (absolute) asynchrony is not generally used as a measure of 'prediction'–in fact, you explicitly cannot tell if the greater asynchrony is in advance of or after the intended target, therefore it specifically obscures predictive vs. reactive taps. Usually, asynchrony is given as a measure of phase accuracy, especially when unsigned. If the authors want to make conclusions about predictive ability specifically, they should report unsigned asynchronies. This leads to my next point.

I think the 4 dependent behavioural variables would benefit from relating to a phase/period framework. Asynchronies index phase accuracy (but of course tend to relate to overall period accuracy, as tapping at an incorrect period would likely lead to being out of phase, but tapping at an incorrect phase does not necessarily lead to tapping at the incorrect period). The 'constant error' measures appear to be a measure of period accuracy that is relatively agnostic to phase, and the standard deviation is measured on reproduced intervals only, therefore also only seems to be a measure of period variability, agnostic to phase.

Moreover, perhaps that could be part of the reason you only see relationships with asynchrony (i.e., that the precision of the alignment between taps and stimuli is what the auditory–motor system is judging, but this same precision relative to an internal clock (which is more what your period measures are)) is not supported by the same system? And that relationship is not embodied in visual–motor synchronization (perhaps because our visual system timing for punctate events is generally poor)? In any case, I think phase/period distinctions would make your measures easier to relate to existing literature (which does often distinguish between phase and period) and also may provide some explanation for the lack of relationship to the brain data for some measures but not others.

---

## [Author Response]

Essential revisions (for the authors):1) The statistical approach will need streamlining and adequate correction for respective families of hypotheses where appropriate. As indicated by the reviews, Having multiple outcome measures also poses a potential risk of inflated type I error. See esp. comments by Rev #1 and Rev #2 in this regard! Sample size justifications and corresponding tempering statements/quantifications about statistical power are probably in order as well.

We addressed these issues with new analysis and a set of precautionary notes in the results and Discussion sections.

2) More background information is needed for the structural MR data and analyses, see esp. Rev # 2.

We carefully respond to Reviewer 2 regarding the structural MR data and analyses.

3) The reviewers' repeated calls for overall editing for clarity, in terms of structure as well as grammar, is not to be taken lightly. Thank you.

The manuscript has been carefully edited by a professional proofreader.

Reviewer #1 (Recommendations for the authors):– By far, one of the most major things to note here is that there are a wide variety of statistical tests carried out across levels. That is, there are 4 behavioral measures – not to mention multiple intervals and modalities and phases, 3 fiber density measures, at 5 depths, as well as other DWI measures tested. Was there any attempt to control for these multiple comparisons? I note that this is different from the within–brain multiple comparisons (i.e. across voxels).

We appreciate and share the concern regarding multiple comparisons. To address this critical issue, we performed a regularized canonical correlation analysis (rCCA) between the behavioral data of the SCT and the structural information of the SWM. This approach allowed us to assess independently the correlation of our AFD measurements of every vertex with every variable of the SCT. The paper has a new segment in the methods describing the analysis and a new section in the results describing the following findings:

“2.4 Canonical correlation between behavioural and AFD maps

In the previous section we correlated many behavioural measures with all vertices of the AFD maps, with the potential risk of inflating the type I error. To address this critical issue, we performed a regularized canonical correlation analysis (rCCA) between the behavioural data of the SCT and the structural information of the SWM (see Methods). This approach allowed us to jointly assess the correlation of our AFD measurements of every vertex with every variable of the SCT. Specifically, rCCA was calculated between the matrix of behavioural parameters from the synchronization phase of the SCT (i.e. absolute asynchrony, constant error, temporal variability, and lag1 autocorrelation) for each sensory modality (auditory and visual), every target interval (550-950) and the AFD matrix of the whole brain. Given the orthogonality between tanAFD and parAFD, and their collinearity with totalAFD, a separate model was built for *totalAFD, _tan_AFD* and *_par_AFD*. Notably, all pairings of behavioural and AFD data rendered highly correlated canonical variates. In line with the previous results, the highest correlation was found between the SCT data and the *_tan_AFD*, followed by *_total_AFD*, and then *_par_AFD* (Figure 6). The loadings of the SCT variables were very consistent across models (Figure 6B) and, invariably, identified the asynchrony of the auditory modality for the 650-850ms intervals as the SCT conditions with larger correlation with the structural data. The correlation between all vertices and their canonical variates was negative, corroborating the hypothesis that subjects with larger predictive abilities had larger *_tan_AFD* in the audiomotor circuit (Figure 6C). A novel result from the rCCA is the significant association between the temporal variability of the auditory phase in the same intervals (650-850 ms) and *_tan_AFD*, although in the opposite direction than the asynchrony (Figure 6B). “

Since the spatial distribution of significant vertices is lost in the rCCA, we have opted to show the main results of our original surface-based linear models (Figures 3-5), with rCCA providing ancillary support to the main findings.

– The second major claim of the paper is the chronotopic organization in the corpus callosum. I find it difficult to assess whether this is true or not. From my observation, there are certainly different parts associated with different intervals, but it does not seem to go in a gradient direction (anterior–posterior, as the authors state). For example, the 750ms interval appears to cover most of the area, and the 950ms interval seems to be in a completely separate area. It's possible that there is a gradient here, but the authors have not convincingly presented it; perhaps by collapsing across dimensions this could be shown, but otherwise, any claims of "chronotopy" similar to what Protopapa and colleagues observed should be tempered.

We agree with the reviewer. We did not test a model with a gradient to determine a chronotopy map similar to what Protopapa and colleagues observed with fMRI. In fact, the longer duration showed significant correlations between the asynchronies in the SCT auditory condition and FDC in the most frontal and occipital section of the CC. Consequently, in the Results section we avoided the use of the concept of chronotopic map and instead used an interval-selectivity map in the CC as follows:

“The association between rhythmic entrainment and white matter properties defined an interval-selectivity map in the CC, with FDC at different levels of CC showing significant correlations with the absolute asynchronies at specific intervals (Figure 7). This interval-selectivity map showed behavioural and structural associations for short and long intervals in the anterior and posterior portions of the CC, respectively. Thus, the FDC fixel values of the posterior midbody of CC (interconnecting motor, pre-motor cortices and M1) showed a significant negative correlation with absolute auditory asynchronies for the 650 and 750 ms intervals (Figure 7BC; FWE-corrected P-value < 0.05). For the asynchronies at the intermediate interval of 850 ms, the negative correlation was observed with FDC fixel values located in the isthmus and the splenium (Figure 7C; interconnecting primary motor, temporal, and visual cortices). Finally, the asynchronies of the interval of 950 ms showed a significant negative correlation with fixels located in forceps minor (FMI) and major (FMA) (Figure 7A,D); interconnecting prefrontal and visual cortices, respectively.”

In addition, in the discussion we tone down our claims of a chronotopic topography. Now the paragraph in the discussion reads:

“Our measurements on the FBA also revealed a topographical arrangement in the correlations between the density of deep fibers and in CC and the rhythmic predictive abilities of subjects, so that significant anatomo-behavioural associations were found in the anterior part of the CC for short intervals and in the posterior CC for long tapping tempos (Schwartze et al., 2012). Nevertheless, our interval-selective map is defined by the correlation between the asynchronies and the FDC, with no topographic model of the distribution of preferred intervals as in the fMRI studies. Moreover, we found a frontal CC cluster of fixels with longer interval selectivity, producing a discontinuity in the anterior-posterior gradient of preferred intervals. These findings suggest that the map for duration selectivity starts anteriorly in the CC linked to the premotor system and ends in the CC of the visual areas of the occipital lobe. In line with our results, a recent imaging study described the existence of large chronomaps covering the cortical mantle from the dorsal and ventral premotor areas to the occipital pole (Harvey et al., 2020; Hendrikx et al., 2022).”

– There is, of course, concern about the sample size in this study. A group of 32 subjects can certainly show associations, but only if they are quite high and potentially subject to bias. This is not meant to say that the results are certainly false, but merely that there is an exploratory aspect to the findings that should be emphasized.

We agree that a larger sample size would lend further support to our findings. The long sessions for psychophysical evaluations and MRI would translate into a time-consuming and expensive endeavor. As the Reviewer notes the reported associations are large, particularly between the asynchronies and the AFP maps, that were specific for the auditory condition in a limited range of metronome tempos (650, 750 ms) (Figure 3- supplement 1). Similar results were obtained with the rCCA where all vertices of the AFD matrix and all the behavioral parameters were included. We included the following phrase in the discussion tempering our findings:

“A potential limitation of our study is the relatively small number of participants relative to the large number of statistical tests performed. To minimize the possibility of statistical errors, we performed a rCCA to jointly model the behavioral and imaging metrics. This provided further support to the importance of superficial cortico-cortical communication through U-fibers and the SCT. While the identified associations are large, and despite our efforts to control for statistical errors, it is inescapable that a sample of 32 subjects with specific inclusion and exclusion criteria may not represent the population. We expect our current findings to be replicated and extended in future studies.”

– The motivation is a bit awkward and amounts to: "we don't know the role of structural connectivity in rhythmic entrainment; we hypothesize that structural connectivity accounts for individual differences in rhythmic entrainment". That is, we don't know if X is involved, therefore we hypothesize that X is involved? Why? While it's true that little work has been done in this specific area, the hypothesis is quite unformed.

We apologize for the lack of clarity. We eliminated the initial statement and now the hypothesis of the paper reads as follows:

“We hypothesize that if the cortical connectivity of the audiomotor system is defining rhythmic entrainment abilities, the individual differences in tapping synchronization should covary with the degree of connectivity of fibers running superficially to the cortical surface and the deep tracts of this system (Assaneo et al., 2019; Steele et al., 2013). In addition, we conjecture that the relationship between the rhythmic tapping abilities and the structural connectivity of the audiomotor system should be more evident for intervals close to the preferred tempo, since previous evidence suggest that this system is tuned at a limited interaction rate (Zalta et al., 2018; Morillon, B., Arnal, L. H., Schroeder, C. E. and Keitel, A. Prominence of δ oscillatory rhythms in the motor cortex and their relevance for auditory and speech perception. Neurosci. Biobehav. Rev. 107, 136– 142 (2019)). “

– The authors should likely read and cite a paper by Zalta, Petkoski, and Morillon (2018) on periodic temporal attention. Here, the authors observed that humans were best at attending to auditory rhythms at a rate of ~1.4Hz, very close to the rate observed in the present study. Moreover, for visual rhythms, the best rate was far slower, at ~0.7Hz. It is possible then that the reason for the lack of effects in the visual condition is that the authors did not test any intervals that were slow enough.Zalta, A., Petkoski, S., and Morillon, B. (2020). Natural rhythms of periodic temporal attention. Nature communications, 11(1), 1–12.

We thank the reviewer for the suggestions of the excellent paper that is really close to our findings. We rewrote the corresponding paragraph in the introduction as follows:

“The existence of neural circuits with preferred intervals conforming chronotopic maps goes in line with the human flexibility to tap in phase (with asynchronies close to zero) and high precision to isochronous stimuli with a wide range of interstimulus-onset intervals (ISI), spanning from 400 to 1200 ms (Mates et al., 1994, Repp 2005). Within this window, subjects show a spontaneous rhythmic tempo, which corresponds to the interval produced naturally when asked to tap in the absence of external cues (McAuley et al., 2006; Zamm et al., 2020). This spontaneous or preferred tempo is around 600 to 750 ms in human adults (Fraisse, 1963, 1978, but see Parncutt, 1994), being faster in early childhood and slower in elderly individuals (McAuley et al., 2006). A recent study demonstrated that the attentional capacities of humans to rhythmic stimuli also have a preferred tempo, with an optimal sampling rate of ~1.4 Hz (ISI of 714 ms) in audition and ~0.7 Hz (ISI of 1428 ms) in vision. Furthermore, motor tapping helps to synchronize the temporal fluctuations of attention with maximal effects at ~1.7 Hz (ISI of 588ms) but only to the auditory modality (Zalta et al., 2018). These findings support the notion that attention and beat perception are shaped by ongoing motor activity, which imposes temporal constraints on the sampling of sensory information within a narrow frequency range. Hence, the audiomotor system is built to optimally work at a preferred tempo.”

Finally, we added this paragraph in the discussion following the referee’s comments:

“The correlations between the density of tangential U-fibers in the right audiomotor circuit and the asynchronies for the auditory condition were interval selective for the intermediate tested tempos. This interval specificity supports the idea that the spontaneous rhythmic tempo observed in many studies of rhythmic entrainment, and with values between 600 and 750 ms, (Drake et al., 2000a,b; McAuley et al., 2006 Bella, S. D. et al. BAASTA: Battery for the assessment of auditory sensorimotor and timing abilities. Behav. Res. Methods. 1128–1145 https://doi.org/10.3758/s13428-016-0773-6 (2017).) (see Figure 2D) has its biological origins in the structural properties of the U-fibers running superficially across the audiomotor circuit. Interestingly, Zalta, et al., (2018) showed that humans were best at attending to auditory rhythms at a rate of ~1.4Hz and that overt motor activity optimizes auditory periodic temporal attention at that same rate. This rate is very close to the intervals with structural associations in the present study. Consequently, we suggest that the cortico-cortical connectivity within the audiomotor system is specially designed to support the internal representation of an auditory pulse at the preferred tempo, providing larger predictive abilities across subjects for the spontaneous motor tempi (Balasubramaniam et al., 2021). On the other hand, Zalta, et al., (2018) also demonstrated that for visual rhythms the best rate was far slower, at ~0.7Hz. Therefore, it is possible that we did not get effects in the visual condition because that we did not test target intervals that were slow enough for this modality, our largest target interval was 950 ms.”

– How was the feedback conveyed? The methods state they are the mean asynchrony, but was this an offset value? A proportion? Did subjects know if they were tapping too fast or too slow?

As stated in the legend of Figure 1 the mean produced and the target interval were displayed at the end of each trial as feedback. Hence, the subjects could know whether their produced intervals were on average shorter or longer than the target interval. We changed the corresponding section in the Methods, since it incorrectly stated that the feedback and the criterion for rejecting incorrect trials was based on the asynchronies. Now the paragraph read as follows:

“A trial was considered correct if, for every interval, the absolute difference between the produced and the target interval was below 30% of the target interval during the synchronization and below 40% during the continuation epoch.”

– There is some concern regarding the latencies for which the tapping information is collected. If the monitor was running at 60Hz, is this the same sample rate for the taps? In this case, there might be a concern that the measures being collected are not sufficiently precise to reveal individual differences, which may explain why there were so few correlations except for asynchrony.

The touch screen was sampled every 1 ms. Hence, we had enough resolution to capture individual differences across all our behavioral parameters.

– What if subjects tapped the monitor but missed the target? Was the tap not collected? Did all subjects remain within the target?

Only correct trials were analyzed. A trial was considered correct if, for every interval, the absolute difference between the produced and the target interval was below 30% of the target interval during the synchronization and below 40% during the continuation epoch.

– More detail about the FWE correction is needed. Was this at the cluster level? Voxel level? The text describes "FWE <0.0.001" and "Cluster<0.001", so what does this mean, precisely? Also, the methods state at one point that a "permutation test" was done, then at another point state random field theory

Correction for multiple comparisons of surface-based analyses was performed with random field theory (RFT). This approach was chosen because data projected on a two-dimensional manifold such as the cortical surface can be spatially smoothed without cross-contamination between adjacent gyri, and therefore lends itself to cluster-wise statistical inference. The vertex-wise cluster forming threshold was p<0.001 (a value recognized to reduce falta associations; Elund PNAS 2016), and a cluster was deemed significant if it had a pcluster<0.001 after RFT. On the other hand, deep white matter was evaluated through fixel-based analysis (FBA) through the tools available in mrtrix 3.0, which includes FWE through permutation testing with connectivity-based fixel enhancement, as described by Raffelt (2015).

– For the continuation phase of the task, did the authors consider decomposing the inter–tap–interval variance using the Wing and Kristofferson (1973) model? This would provide measures of putative "central" and "motor" variance. Given that multiple intervals were tested, the authors should observe that central variance grows with the interval, whereas motor variance remains flat. Further, one might expect motor variance to relate more to connections within motor regions, whereas central variance could relate to audiomotor connections.

Unfortunately, the number of produced intervals in the continuation phase is six, which is too small to generate stable estimates of the central and motor variance using the Wing and Kristofferson (1973) model.

– For continuation performance, a commonly observed phenomenon is "drift" in which subjects continuously speed up or slow down their responses. Was this observed at all? A typical method for drift correction is to detrend the time series with a first–order polynomial and use the residuals for calculating measures of variance (Vorberg and Wing, 1996). Was this examined at all?

The drift that we obtained in the continuation phase was small, due to the small number of produced intervals in the epoch of the task. The detrended data showed very similar effects on modality and target duration and are not reported in the paper.

Supplementary figure 11 shows that around 20% of trials showed positive lag 1 autocorrelations across interval duration and modality conditions. As suggested by the reviewer our data show some violations to the error correction notion.

– Another prediction of the W–K model is the negative lag 1 autocorrelation which the authors examine here. Yet, violations of this commonly do occur. Were there violations (i.e. trials) where subjects did not show negative autocorrelations?Reviewer #2 (Recommendations for the authors):The selection of interval ranges needs further justification–why did the researchers choose these particular interval lengths, and did they predict differences between them? The preferred tempo range includes the 650 ms interval, as they note, but most authors would suggest that the 550 ms interval rate would be in this range for many people as well, as preferred beat rates tend to span 300–900 ms (Parncutt, 1994). Thus, was the range of intervals selected to provide evidence of structural chronomaps in the brain, or did the researchers expect to see behavioural differences across the intervals (that may be related to structural brain measures) that they felt would be meaningful (being more or less related to optimal tapping rates), and therefore they used a range of intervals to assess this?

We agree with the reviewer. Therefore, in the introduction we added first this paragraph:

“The existence of neural circuits with preferred intervals conforming chronotopic maps goes in line with the human flexibility to tap in phase (small asynchronies) and high precision to isochronous stimuli with a wide range of inter-stimulus onset intervals (ISI), spanning from 400 to 1200 ms (Mates et al., 1994, Repp 2005). Within this window, subjects show a spontaneous rhythmic tempo, which corresponds to the interval produced naturally when asked to tap in the absence of external cues (McAuley et al., 2006; Zamm et al., 2020). This spontaneous or preferred tempo is around 600 to 750 ms in human adults (Fraisse, 1963, 1978, but see Parncutt, 1994), being faster in early childhood and slower in elderly individuals (McAuley et al., 2006). A recent study demonstrated that the attentional capacities of humans to rhythmic stimuli are limited, with an optimal sampling rate of ~1.4 Hz (ISI of 714 ms) in audition and ~0.7 Hz (ISI of 1428 ms) in vision. Furthermore, motor tapping helps to synchronize the temporal

fluctuations of attention with maximal effects at ~1.7 Hz (ISI of 588ms) (Zalta et al., 2018). These findings support the notion that attention and beat perception are shaped by ongoing motor activity, which imposes temporal constraints on the sampling of sensory information within a narrow frequency range. Hence, the audiomotor system is built to optimally work at a preferred tempo.” Then late in the introduction we added this phrase:

“We hypothesize that if the cortical connectivity of the audiomotor system is defining rhythmic entrainment abilities, the individual differences in tapping synchronization should covary with the degree of connectivity of fibers running superficially to the cortical surface and the deep tracts of this system (Assaneo et al., 2019; Steele et al., 2013). In addition, we conjecture that the relationship between the rhythmic tapping abilities and the structural connectivity of the audiomotor system should be more evident for intervals close to the preferred tempo, since previous evidence suggest that this system is tuned at a limited interaction rate (Zalta et al., 2018; Morillon, B., Arnal, L. H., Schroeder, C. E. and Keitel, A. Prominence of δ oscillatory rhythms in the motor cortex and their relevance for auditory and speech perception. Neurosci. Biobehav. Rev. 107, 136– 142 (2019)).

Finally, we added the sentence in the last paragraph of the introduction:

“We used an ISI range of 550-950 ms since it contains the preferred interval and is within the optimal window for tap synchronization (Repp 2005). Hence, with this ISI range we could potentially identify structural correlates for both the preferred tempo and interval selectivity.”

There are 2 modalities and 5 different interval lengths, all of which appear to be analyzed independently in the brain data. The MRI methods contain very little information about multiple comparisons correction related to the number of different tests that are performed–this needs much clearer explication.

This is a very important point, which was also raised by Reviewer 1. To minimize the possibility of false positive findings, we performed a rCCA analysis to jointly model the behavioral metrics on one hand, and AFD metrics of all vertices throughout the surface, on the other. Further details are provided in the response to the first point by Reviewer 1, and this new analysis is described in the Methods.

Unsigned (absolute) asynchrony is not generally used as a measure of 'prediction'–in fact, you explicitly cannot tell if the greater asynchrony is in advance of or after the intended target, therefore it specifically obscures predictive vs. reactive taps. Usually, asynchrony is given as a measure of phase accuracy, especially when unsigned. If the authors want to make conclusions about predictive ability specifically, they should report unsigned asynchronies. This leads to my next point.

Based on the excellent comments of the reviewer we adopted a phase/period framework for the analysis and interpretation of the behavioral parameters of the SCT task. We performed the correlations between the signed asynchronies and the superficial and white matter maps, with very limited effects. Therefore, we dropped the framework of prediction abilities and replaced it with the notion of phase accuracy, as suggested by the referee in the next comment.

I think the 4 dependent behavioural variables would benefit from relating to a phase/period framework. Asynchronies index phase accuracy (but of course tend to relate to overall period accuracy, as tapping at an incorrect period would likely lead to being out of phase, but tapping at an incorrect phase does not necessarily lead to tapping at the incorrect period). The 'constant error' measures appear to be a measure of period accuracy that is relatively agnostic to phase, and the standard deviation is measured on reproduced intervals only, therefore also only seems to be a measure of period variability, agnostic to phase.

Now the paper uses a phase/period framework, where the asynchronies are an index phase accuracy, the constant error is a measure of period accuracy, and the temporal variability is a measure of period variability.

In the introduction we describe this framework as follows:

“SCT rhythmic performance across durations (ISI: 550,650,750,850 or 950 ms) and modalities (auditory and visual) was characterized using the absolute asynchronies, the autocorrelation of the inter-tap interval time series during the synchronization epoch, the constant error, and the temporal variability during both synchronization and continuation epochs. These parameters measure the phase accuracy, error correction, period accuracy, and period precision of the rhythmic tapping of the subjects, respectively (Figure 2A).”

Moreover, perhaps that could be part of the reason you only see relationships with asynchrony (i.e., that the precision of the alignment between taps and stimuli is what the auditory–motor system is judging, but this same precision relative to an internal clock which is more what your period measures are) is not supported by the same system? And that relationship is not embodied in visual–motor synchronization (perhaps because our visual system timing for punctate events is generally poor)? In any case, I think phase/period distinctions would make your measures easier to relate to existing literature (which does often distinguish between phase and period) and also may provide some explanation for the lack of relationship to the brain data for some measures but not others.

We fully agree with the reviewer and adopted the phase and period framework throughout the paper.